# Advances in the Application of Graphene and Its Derivatives in Drug Delivery Systems

**DOI:** 10.3390/ph18091245

**Published:** 2025-08-22

**Authors:** Changzhou Jin, Huishan Zheng, Jianmin Chen

**Affiliations:** 1School of Environmental and Energy Engineering, Anhui Jianzhu University, Hefei 230601, China; 2School of Pharmacy and Medical Technology, Putian University, Putian 351100, China; 3Key Laboratory of Pharmaceutical Analysis and Laboratory Medicine (Putian University), Fujian Province University, Putian 351100, China

**Keywords:** graphene, nanomaterials, drug carrier, targeting

## Abstract

Graphene, owing to its exceptionally high specific surface area, abundant surface functional groups, and outstanding biocompatibility, exhibits tremendous potential in the development of nanodrug delivery systems. This review systematically outlines the latest research advancements regarding graphene and its derivatives in drug loading, targeted delivery, and smart release. It covers delivery strategies and mechanisms for various types of drugs, including small molecules and macromolecules, with a particular emphasis on their applications in major diseases such as cancer, neurological disorders, and infection control. The article also discusses stimulus-responsive release mechanisms, such as pH-responsiveness and photothermal responsiveness, and highlights the critical role of surface functionalization of graphene and its derivatives in enhancing therapeutic efficacy while reducing systemic toxicity. Furthermore, the review evaluates key challenges to the clinical translation of graphene-based materials, including safety, toxicity, and metabolic uncertainties. It points out that future research should focus on integrating structural modulation of materials with biological behavior to construct intelligent nanoplatforms featuring biodegradability, low immunogenicity, and precise therapeutic targeting. The aim of this paper is to provide theoretical insights and technical guidance for the customized design and precision medicine applications of graphene and its derivative-based drug delivery systems.

## 1. Introduction

As the most abundant non-metallic element on Earth, carbon is not only a fundamental unit of biological systems but also a crucial cornerstone in modern materials science. Among its various allotropes, graphene has emerged as a focal point in nanomaterials research due to its unique two-dimensional structure and outstanding physicochemical properties. Graphene consists of a single layer of *sp^2^*-hybridized carbon atoms arranged in a hexagonal honeycomb lattice, with an ultra-thin atomic layer thickness of approximately 0.34 nm and a highly ordered crystalline structure. This architecture endows graphene with extraordinary mechanical strength, including a Young’s modulus up to 1 TPa and fracture strength of 130 GPa. It also exhibits exceptional electron mobility, reaching 200,000 cm^2^/V·s—far surpassing conventional materials. Its high specific surface area (up to 2630 m^2^/g) and abundant surface chemical sites make graphene an ideal platform for functional modification. In addition, its superior electrical conductivity, thermal conductivity, and mechanical strength support its broad application prospects in areas such as energy storage, sensors, and biosensing [1,2,3,4]. In most applications, graphene appears in the form of graphene-based nanostructures (Graphene-Based Nanomaterials, GBNs). This allows for enhanced utilization of its physical and chemical properties. Types of GBNs include graphene oxide (GO), reduced graphene oxide (rGO), graphene quantum dots (GQDs), as well as composite structures formed by graphene with inorganic materials, polymers, or organic nanomaterials [5,6].

In the medical field, graphene’s high surface area and excellent biocompatibility [7] make it an ideal drug carrier, enabling precise control of drug release through surface functionalization, thereby enhancing therapeutic efficacy and reducing side effects. Particularly in anticancer therapy, gene delivery, and targeted treatment, GBNs have demonstrated tremendous potential. By introducing functional groups such as hydroxyl, carboxyl, and epoxy, it is possible to effectively regulate their hydrophilicity/hydrophobicity, zeta potential, and drug-loading capacity, allowing adaptation to various therapeutic needs. Moreover, GO and its oxidized derivatives (e.g., rGO) have shown great promise in multimodal tumor diagnosis and therapy systems. Graphene-based drug delivery platforms achieve both high-efficiency drug loading and sustained release of chemotherapeutic agents—such as paclitaxel (PTX) and doxorubicin (DOX). They also combine photothermal and magnetothermal effects to enable synergistic chemo-thermal therapy, significantly enhancing antitumor efficacy. Drug delivery systems based on GBNs have made remarkable progress. In particular, they improve drug bioavailability, targeting specificity, and controlled release performance [8]. With their high specific surface area, multifunctional surfaces, and excellent biocompatibility, GBNs have become a major research focus in the field of nanodrug delivery systems.

Due to the growing attention on GBNs in the field of drug delivery, the number of related publications has been continuously increasing. According to statistics (Figure 1A), research in this area has entered a rapid development phase since 2014. The annual number of publications has risen exponentially. This growth peaked between 2019 and 2021, during which more than 600 papers were published annually on average. From the perspective of annual publication proportion, studies published between 2018 and 2023 account for the majority of the total literature, with 2019 contributing the highest share at 12.4%. This indicates that the period marks a concentrated surge in research activity on graphene-based drug delivery systems.

Although graphene and its derivatives (Graphene-Family Nanomaterials, GFNs) show great potential in drug delivery, their clinical translation still faces significant challenges. These include biosafety, toxicity mechanisms, and long-term metabolic behavior, all of which remain largely uncertain. Existing studies have shown that graphene materials can exhibit favorable biocompatibility under specific conditions. However, under varying doses, particle sizes, surface modifications, and exposure durations, they may induce cytotoxicity, immune responses, or organ accumulation. This highlights the urgent need for systematic evaluation. The metabolic and degradative pathways of graphene within the body remain unclear, and its long-term retention may induce adverse effects, including chronic inflammation, fibrosis, and even carcinogenesis. Moreover, the mechanisms by which functional modifications influence the biological behavior of graphene are not yet fully elucidated. Factors such as the type, density, and spatial distribution of functional groups may significantly affect their blood circulation dynamics, tissue targeting capability, and clearance processes. Against this backdrop, there is an urgent need to investigate the relationship between graphene’s physicochemical properties and its biological responses, starting from material structure. In particular, A deeper understanding is needed of interaction mechanisms in the delivery of small molecules, macromolecules, and nucleic acid therapeutics. This will guide the rational design of safe and efficient GFN-based delivery systems.

Based on the above background, this review aims to systematically summarize the research progress of GFNs in drug delivery systems, with a particular focus on their loading mechanisms, targeting strategies, and controlled release methods in the delivery of small molecules, macromolecules, and nucleic acid-based drugs. In Figure 1B, Influential publications such as The Rise of Graphene (2007), Graphene and Graphene Oxide as New Nanocarriers for Drug Delivery Applications (2013), and other high-impact studies published through 2025 have had a profound influence on subsequent research directions (Figure 1C). Figure 1B,C further illustrate the intellectual lineage and thematic evolution among publications across different years, demonstrating a shift from early focus on material synthesis to more application-oriented studies such as biocompatibility evaluation and targeted drug delivery.

We begin by examining the relationship between material structure and biological response. From the core literature highlighted in Figure 1B,C, we extract key insights and summarize recent representative research achievements. We analyze how different material configurations affect delivery performance and in vivo behavior. Special emphasis is placed on their advantages and limitations in treating cancer, neurodegenerative diseases, and infectious diseases. This analysis aims to provide a theoretical foundation and practical guidance for the safe design and clinical translation of future GFN-based drug delivery systems.

## 2. Preparation, Drug Loading, and Targeted Delivery Strategies of GFNs

### 2.1. Preparation Methods of GFNs

Over more than a century of technological evolution, graphene synthesis methods have become increasingly diverse, resulting in a variety of distinct fabrication strategies. Currently, commonly used graphene preparation techniques include chemical oxidation (e.g., the Hummers method), mechanical exfoliation, liquid-phase exfoliation, electrochemical exfoliation, and chemical vapor deposition (CVD). Among these, GO is the most widely used form of GFNs. Its synthesis has evolved from the Brodie and Staudenmaier methods to the widely adopted Hummers method [10]. These methods have become standardized at both laboratory and industrial levels. They continue to be optimized to meet demands in quality, cost, and scalability.

#### 2.1.1. Mechanical Methods for Graphene Synthesis

Mechanical exfoliation is one of the earliest classical methods used to isolate monolayer graphene, based on the principle of physically peeling off graphene layers from bulk graphite. This technique was first applied by Geim and Novoselov [11] in 2004 using highly oriented pyrolytic graphite (HOPG). By repeatedly peeling the graphite with ordinary adhesive tape, they successfully obtained high-quality monolayer graphene. In earlier work, Lu et al. used an atomic force microscope (AFM) probe to peel multilayer graphite flakes from graphite, with a thickness of approximately 200 nm, laying the groundwork for the development of subsequent mechanical exfoliation methods [12].

Specific forms of mechanical exfoliation include adhesive tape, AFM probes, ultrasonic agitation [13], and ultra-sharp diamond wedge tools. These apply directional mechanical force to cleave graphite. These methods can produce high-purity, low-defect, and structurally intact graphene sheets, making them particularly suitable for fundamental research and device development. For example, the diamond wedge-based exfoliation technique developed by Jayasena and Subbiah, when combined with ultrasonic vibration, enables the efficient isolation of few-layer graphene from HOPG [14]. As shown in Figure 2, a piece of HOPG with dimensions of 2 × 12 × 12 mm is first cut into smaller blocks of 1 × 1 × 2 mm. These blocks are embedded in resin for stabilization and then shaped into a pyramidal geometry to facilitate the subsequent exfoliation process. The exfoliation tool consists of an ultra-sharp single-crystal diamond wedge with an included angle of 35° and a tip sharpness of less than 20 Å. This wedge tool is mounted on an adjustable-frequency ultrasonic oscillator (operating between 25 and 45 kHz) with a vibration amplitude on the order of tens of nanometers. Control is achieved by tuning the voltage between 0 and 30 V. During operation, the diamond wedge and HOPG sample are precisely mounted and aligned on a high-precision sliding rail system. The HOPG is moved downward at a speed of 0.6 mm/s, allowing its surface to overlap the wedge tool by approximately 40 nm, while the wedge remains stationary. During exfoliation, graphene layers slide along the wedge surface and float onto the water below. They are then captured with a collection ring and transferred onto copper grids or Si/SiO_2_ substrates for AFM, transmission electron microscopy (TEM), and Raman spectroscopy. Typically, the experiment involves 20 exfoliation cycles, with one graphene layer produced per cycle, ultimately yielding graphene sheets with dimensions of approximately 900 × 300 μm and a thickness of several tens of nanometers. Experimental results show that ultrasonic vibration improves exfoliation efficiency. It also reduces defect density and grain boundaries, thereby improving the graphene’s crystallinity and structural integrity.

The greatest advantage of mechanical exfoliation lies in the exceptionally high quality of the resulting graphene. It features an intact lattice structure with minimal defects, outstanding electron mobility, and superior mechanical properties. These qualities make it ideal for studying the intrinsic physical properties of graphene. However, mechanical exfoliation also presents significant limitations, including very low yield, poor process reproducibility, reliance on manual operation, and unsuitability for large-scale production. As a result, this method is currently primarily used in laboratory settings to obtain high-quality graphene samples, rather than for the industrial synthesis of GO [15].

#### 2.1.2. CVD Method

Although mechanical exfoliation of graphite crystals can yield high-quality graphene at the microscale, this method is not suitable for large-scale production [11]. As a result, researchers have explored several alternative synthesis techniques. These include chemical exfoliation of bulk graphite, epitaxial growth on SiC substrates, and chemical vapor deposition (CVD) on metal substrates like Ni or Cu. Currently, CVD has proven to be the most successful method for producing large-area, high-quality graphene films, with thicknesses down to the nanometer scale. The CVD process typically involves a catalytic reaction between hydrocarbon gases and a metal catalyst substrate. The initial CVD synthesis of graphene was achieved on a Ni substrate, typically conducted under low-pressure conditions at around 1000 °C. During the process, carbon atoms generated from the decomposition of carbon source gases dissolve into the Ni substrate, and subsequently precipitate and crystallize into graphene during the cooling phase. In contrast, the growth mechanism on Cu substrates is fundamentally different. Due to the negligible solubility of carbon in Cu, carbon atoms adsorb and diffuse only on the surface, eventually forming an ordered hexagonal lattice structure. Therefore, the use of commercial Cu foil enables the synthesis of primarily monolayer, large-area single-crystal graphene under economically viable and controllable conditions, paving the way for its practical applications in areas such as large-area transparent electronic conductors. Based on this growth mechanism, the key to achieving large-area single-crystal graphene lies in reducing the nucleation rate and minimizing grain density, thereby ensuring the formation of only a few—ideally a single—graphene crystal.

Copper-based CVD synthesis is a reliable method for producing high-quality graphene films with controllable thickness [16,17,18]. Due to the extremely low solubility of carbon in copper, graphene growth is confined to the copper surface and proceeds via a seed-induced two-dimensional growth process [19]. As illustrated in Figure 3, during the CVD growth process, hydrocarbons (e.g., methane, CH_4_) are converted into graphene through the following basic steps:Adsorption of methane onto the copper surface;Partial dehydrogenation of methane to form carbon-containing species, such as CH_x_ (x = 0–3);Surface diffusion of these carbon species on the copper substrate;Nucleation of graphene at active sites on the copper surface, forming graphene nuclei;Adsorption of carbon species at the edges of graphene nuclei, promoting grain growth;Coalescence of adjacent graphene grains, resulting in the formation of polycrystalline graphene films [20,21,22,23,24].

The main advantage of the CVD method is its ability to control the number of graphene layers while maintaining lattice integrity. This results in graphene with excellent continuity and flatness, making it ideal for high-end applications such as electronic devices, flexible displays, and sensors. However, upon cooling from the high synthesis temperatures to room temperature, CVD-grown graphene exhibits negative thermal expansion, and lateral contraction of the metal substrate can lead to the formation of nanoscale particles. Additionally, the inherent surface roughness and step edges of the Cu substrate can induce sub-nanometer ripples in the graphene structure. These ripples, caused by thermodynamic or morphological factors, can significantly impact the electrical properties of graphene. On the other hand, CVD-grown graphene often exhibits polycrystallinity, a result of graphene grains growing in random directions after nucleation. As a result, its grain boundaries differ from those of HOPG, often containing non-hexagonal carbon rings, vacancies, and other structural defects. These structural defects, including ripples and grain boundaries, show enhanced chemical reactivity [25].Therefore, they must be carefully considered when applying CVD graphene to biointerfaces or functional platforms [26].

#### 2.1.3. Hummer’s Method for GO Preparation

In 1958, Hummers and Offeman proposed a milder and more operable synthesis method, now widely known as the Hummers method, which utilizes a combination of H_2_SO_4_, NaNO_3_, and KMnO_4_ to oxidize graphite. Figure 4 illustrates the detailed procedure: First, under ice bath conditions, graphite powder (1.0 g) is added to a mixed solution composed of concentrated H_2_SO_4_ (23.0 mL) and NaNO_3_ (0.5 g), followed by continuous stirring. Then, KMnO_4_ (3.0 g) is slowly added while strictly controlling the reaction temperature to not exceed 20 °C to prevent a violent reaction. Once the addition is complete, the ice bath is removed, and the reaction mixture is heated to 35 °C and maintained for 30 min to allow the oxidation process to proceed further. Subsequently, distilled water (46 mL) is added slowly, triggering a strong exothermic reaction. The temperature rises to ~98 °C and is maintained for 15 min, during which the mixture turns brown-gray. After cooling, a large volume of water (140 mL) and hydrogen peroxide (H_2_O_2_, 3 mL) is added to terminate the reaction and reduce manganese oxides, turning the system yellow-brown. The resulting product is washed repeatedly by centrifugation until the supernatant reaches neutral pH, then dried by vacuum rotary evaporation at 50 °C, yielding brown graphene oxide, with a final yield of approximately 0.82 g. Although the Hummers method significantly improved the controllability of the synthesis process, it still presents several limitations, including the release of toxic gases, complex operation, low yield, and a tedious purification process. To address these drawbacks, subsequent researchers have proposed various optimizations to the original method. For example, the Tour group eliminated NaNO_3_ to reduce the risk of toxic byproducts [27], Nishina et al. developed a continuous-flow reaction system to accommodate large-scale production [28], and Xing’s team replaced KMnO_4_ with K_2_FeO_4_, enabling the oxidation reaction to proceed at room temperature [29].

### 2.2. Drug Loading of GFNs

With the continuous advancement of nanomedicine, drug delivery systems are rapidly evolving toward precision and intelligence. In this context, the structural tunability and surface functional diversity of carrier materials have become key factors in enhancing delivery efficiency. Graphene-based materials, owing to their excellent water dispersibility, biocompatibility, and rich interaction mechanisms with drug molecules, exhibit outstanding drug-loading potential [30]. The binding modes between graphene-based materials and drug molecules are influenced by the structural characteristics of the drugs, primarily involving:Non-covalent interactions, such as hydrogen bonding, π–π stacking, hydrophobic interactions, and electrostatic adsorption [31,32].Covalent linkages, including amide bonds, ester bonds, and phosphoester bonds [33].

Hydrophobic small-molecule drugs typically bind to the aromatic carbon backbone of graphene-based materials through π–π stacking interactions, stabilized further by hydrophobic forces and hydrogen bonding. This binding strategy forms stable complexes and preserves drug structure. It avoids the disruption typically caused by chemical modifications [34]. To enhance drug loading efficiency and release stability, physical methods such as ultrasonication, stirring incubation, and emulsion crosslinking are often employed to construct composite delivery systems. In addition, surface functionalization is commonly used to improve dispersibility and targeting ability. In contrast, hydrophilic small-molecule drugs primarily rely on hydrogen bonding and electrostatic adsorption for loading. This is particularly relevant for positively charged drugs such as cationic antibiotics, which can interact with carboxyl groups on the surface of graphene-based materials via electrostatic attraction [35,36]. Furthermore, if the drug possesses reactive functional groups, covalent conjugation can be employed to enhance binding strength and controlled release performance. Macromolecular drugs such as proteins, peptides, and nucleic acids possess complex structures and rich distributions of functional groups, often exhibiting multi-point cooperative binding [8]. Their interaction mechanisms commonly involve hydrogen bonding, electrostatic interactions, π–π stacking, metal ion–induced self-assembly, and intercalation-based adsorption between graphene layers. By adjusting material structure and functionalization, adsorptive or covalent strategies can be chosen based on drug polarity, molecular size, and targeting needs. This enables the precise delivery of a wide range of drugs—from small-molecule chemotherapeutics (such as DOX, PTX, and gefitinib), to hydrophilic drugs (such as gentamicin and dexamethasone), and further to macromolecular biologics (including proteins, DNA, and siRNA). The diverse configurations of graphene-based materials support a variety of delivery strategies, offering a strong foundation for developing personalized, highly targeted smart drug delivery systems. We summarized the ten most frequently utilized drugs in graphene-based drug delivery systems and their respective proportions, as illustrated in Figure 5. 

#### 2.2.1. Delivery of Hydrophobic Small-Molecule Drugs

Hydrophobic small-molecule drugs, due to their poor water solubility and tendency to aggregate in biological fluids without specific distribution, often suffer from low bioavailability, poor targeting, and high systemic toxicity, which limits their widespread clinical application [31,37]. GFNs, with their hydrophobic carbon backbone and tunable surface structures, offer distinct advantages in improving the delivery of hydrophobic drugs. Their amphiphilic two-dimensional layered structure provides an excellent adsorption platform, enhancing the drug’s dispersion in aqueous environments, preventing early precipitation and nonspecific interactions, and thereby improving drug stability and bioavailability. In addition, GFNs support uniform surface loading, helping to minimize dosage fluctuations and reduce adverse side effects. Through further structural engineering and surface functionalization, these carriers can also regulate drug release behaviors under different physiological conditions, enabling more precise and sustained therapeutic effects [38,39]. To date, graphene-based nanoplatforms have been widely used in the development of delivery systems for various hydrophobic anticancer drugs, such as PTX, DOX, and cisplatin, effectively optimizing their in vivo pharmacokinetics and targeted therapeutic performance.

In most graphene-based drug delivery systems, researchers often combine multiple physical processes and surface modification strategies to construct composite delivery platforms. Specifically, commonly used physical methods include: High-intensity ultrasonication to promote drug intercalation or to induce hollow structure formation in the material [40]; Stirring incubation, which allows drugs to adsorb effectively onto the material surface in aqueous or organic solvents [41]; Emulsion crosslinking or solvent evaporation methods, frequently used to create microcapsule structures for controlled release [42]. These physical techniques are often integrated with self-assembly strategies. For example, Liquid ammonia treatment can induce the formation of hollow microspheres [43]; Interfacial adsorption at oil–water interfaces can enhance stable drug encapsulation [44]. Together, these approaches provide multifunctional control over drug loading and release behavior, particularly suited to complex therapeutic requirements.

Graphene and its derivatives can also be significantly enhanced in drug affinity and environmental responsiveness through surface chemical modification. For example, using EDC/NHS coupling reactions to graft hydrophilic or targeting functional molecules such as hyaluronic acid (HA) or polyethylene glycol (PEG) can improve circulatory stability and targeting recognition ability in vivo [45]. Furthermore, the platform can be functionalized with biorecognition units such as antibodies or aptamers to enable specific localization to target tissues [46]. In addition, by incorporating charge modulation strategies or introducing pH-responsive functional groups, graphene-based carriers can be endowed with stimuli-responsive release capabilities, allowing them to respond selectively to the pathological microenvironment [9].

Overall, the interaction between GFNs and hydrophobic drugs primarily relies on non-covalent adsorption mechanisms. Building upon this foundation, the integration of physical techniques—such as ultrasonication, stirring incubation, and interfacial assembly—combined with surface functionalization strategies, not only enhances drug-loading capacity, but also offers diverse approaches for controlled release and targeted therapy. The following section will explore, using representative drugs as examples, the specific binding mechanisms between these agents and GFNs, along with a comprehensive evaluation of their pharmacological effects.

The loading of hydrophobic drugs onto GO-based materials typically relies on π–π stacking interactions mediated by the aromatic ring structures within the drug molecules. This process is often accompanied by hydrophobic interactions and hydrogen bonding, resulting in the formation of stable non-covalent complexes. For example, a variety of drugs including paclitaxel (PTX) [43,47], doxorubicin (DOX) [48], rifampicin (RFP) [49], quercetin (QSR) [50], camptothecin (CPT) [51], imatinib [52] and ganoderic acid D (GAD) [53] all contain aromatic structures within their molecular frameworks, enabling them to effectively undergo π–π stacking adsorption onto the GO substrate. Among these, DOX and PTX have polar groups (e.g., phenolic hydroxyl, amino, ester) that form hydrogen bonds with GO’s surface groups, enhancing drug immobilization. In contrast, RFP, imatinib, and GAD primarily exhibit hydrophobic interactions with GO, forming more stable complexes particularly under anhydrous or organic phase conditions, demonstrating higher drug-loading affinity [49,53].

Building upon these mechanisms, some studies have further introduced covalent modification strategies. For example, PTX or DOX can be pre-functionalized with carboxyl or amino groups and then covalently coupled to GO via amide bond formation. This enhances drug-binding stability and improves controlled release. In particular, DOX is often designed as a pH-responsive covalently linked system [48], where the release rate accelerates under the slightly acidic tumor microenvironment, enabling precise release control. Similarly, QSR has been engineered into systems combining π–π stacking with photosensitive response. This enables intelligent, stimuli-responsive drug release under specific environmental triggers [54].

In summary, aromatic structures, hydrophobicity, and polar functional groups play crucial roles in mediating interactions between hydrophobic drugs and GO. The choice and optimization of binding modes critically influence the stability of the drug carrier system, as well as its controlled-release behavior and targeting efficiency.

#### 2.2.2. Delivery of Hydrophilic Small-Molecule Drugs

Compared to hydrophobic drugs, hydrophilic molecules tend to diffuse, degrade, or clear quickly in vivo. This is due to their high water solubility and polar groups, which reduces stability and therapeutic efficacy. The high specific surface area and abundant surface oxygen-containing functional groups of GFNs enable them to form stable delivery complexes with hydrophilic drugs through multiple mechanisms, including hydrogen bonding, electrostatic adsorption, or covalent bonding. This effectively enhances the in vivo stability, controlled-release capability, and targeting efficiency of hydrophilic drugs [55].

In constructing GFNs-based delivery systems for hydrophilic drugs, researchers commonly employ strategies such as physical adsorption, covalent bonding, and composite material encapsulation to achieve stable drug loading and controlled release [56,57,58]. Due to their high solubility, hydrophilic drugs typically interact with GFNs through mechanisms including electrostatic interactions, hydrogen bonding, and molecular compatibility. These interactions are often complemented by a range of material preparation techniques aimed at improving delivery efficiency and biocompatibility. Among these, ultrasonication and stirring incubation are the most commonly used physical methods, facilitating sufficient contact and binding between the drug molecules and GFN sheets [59]. For systems needing more controlled release, chemical agents like EDC/NHS or DCC/DMAP are used to form covalent bonds between GFNs and drug molecules or polymers (e.g., chitosan, PEG). This improves sustained release and biocompatibility.

In addition to single-material modifications, GFNs are also commonly used to construct composite carriers, such as nanofiber membranes, hydrogels, and porous microsphere systems [60,61,62]. By employing techniques such as electrospinning and emulsion polymerization, GFNs can be combined with polyvinyl alcohol (PVA), chitosan, or alginate, which not only enhances drug-loading capacity but also improves localized drug release characteristics and tissue compatibility.

Due to their abundant polar functional groups, hydrophilic drugs can readily form non-covalent interactions with the GO surface through mechanisms such as hydrogen bonding and electrostatic interactions, thereby achieving stable loading and controlled release [63].

For example, aminoglycoside antibiotics represented by gentamicin, which are rich in amino and hydroxyl functional groups, can firmly adsorb onto the GO surface, demonstrating excellent sustained-release properties [64,65]. Ciprofloxacin (CIP) possesses a molecular structure that features both aromatic rings and carboxyl groups, enabling it to form stable complexes with GO through a combination of π–π stacking and hydrogen bonding mechanisms. This dual interaction enhances both its tissue penetration and local sustained-release properties [66,67]. In certain porous GO nanostructures, CIP can also achieve structural encapsulation through physical intercalation, further improving its targeted delivery efficiency and pharmacological effect retention [67].

In addition to the typical antimicrobial drugs mentioned above, certain hydrophilic small molecules such as metformin, (dexamethasone)dexamethasone, and methotrexate can also form stable complexes with GO through mechanisms including hydrogen bonding, electrostatic adsorption, or covalent modification. For example, the guanidine and amino groups in metformin can form amide bonds with carboxyl groups on the GO surface via EDC/NHS coupling reactions. Alternatively, metformin can be adsorbed through hydrogen bonding onto PEG-functionalized or nitrogen-doped GO quantum dots, achieving sustained-release effects [68]. Both DEX and methotrexate (MTX) contain multiple hydroxyl groups, keto carbonyl groups, or aromatic ring structures in their molecular frameworks, enabling them to form hydrogen-bonding networks and electrostatic interactions with the carboxyl or hydroxyl groups on the GO surface. In some systems, binding can be enhanced by incorporating polysaccharide matrices like chitosan or sodium alginate. This improves both drug-loading and controlled-release performance [69,70,71,72].

Overall, the binding of hydrophilic drugs to GO-based carriers is primarily governed by polar interactions, supplemented by π–π stacking and covalent modifications. Through rational design of material structure and interfacial chemistry, the delivery efficiency and therapeutic targeting of hydrophilic drugs can be significantly enhanced in complex physiological environments such as infections and tumors.

#### 2.2.3. Delivery of Macromolecular Drugs

GBNs exhibit significant advantages in the delivery of macromolecular drugs, particularly for biologics such as proteins, peptides, and nucleic acids, which possess complex molecular structures and poor stability. In research, delivery systems are often constructed in combination with aptamers to achieve targeted recognition and efficient loading [73]. In such systems, aptamers can serve both as recognition modules and as direct delivery cargos, forming multifunctional composite platforms with GBNs [74].

In terms of processing methods, non-covalent adsorption between GBNs and macromolecular drugs is commonly achieved through techniques such as direct mixing, ultrasonication, and stirring incubation. For example, positively charged siRNA or aptamers can be dissolved in buffer and co-incubated with GBNs, enabling electrostatically induced self-assembly adsorption [75]. To further enhance structural stability and delivery performance, GBNs can also be blended with polymers such as PVA or chitosan to prepare delivery systems including nanofiber membranes, hydrogels, or emulsion microspheres [76,77]. These platforms are widely applied in controlled-release oral and injectable drug delivery strategies.

For systems requiring high long-term stability and precise control, covalent modification of GBNs is particularly critical. By chemically coupling drugs or recognition elements to the surface of GBNs, these components can be stably anchored, preventing desorption during the delivery process and enhancing both recognition specificity and therapeutic targeting [78]. At present, this strategy has been widely applied in the construction of targeted delivery systems for insulin, siRNA, and aptamers, demonstrating favorable in vitro and in vivo controlled-release performance as well as therapeutic efficacy [79].

Taking insulin as an example, its molecular structure contains a large number of polar residues [80] (such as carboxyl, amino, and hydroxyl groups) along with several aromatic hydrophobic residues. These features enable insulin to form stable complexes with GO through hydrogen bonding [81], electrostatic adsorption [82] and π–π stacking [81]. Additionally, insulin molecules feature multiple positively charged residues such as lysine on their surface, which can further engage in electrostatic interactions with the carboxyl groups on GO, enhancing the stability of the composite system. Furthermore, GO can also be combined with magnetic or conductive materials to create intercalated carriers. These support nanopore adsorption and enable stimulus-responsive insulin release [83,84]. In some studies, the GO surface is carboxylated and then covalently coupled with insulin or its recognition aptamers via EDC/NHS chemistry, achieving stable drug loading and targeted release [85].

Specifically, antibody drugs such as Trastuzumab are high-molecular-weight immunoglobulins containing abundant aromatic amino acids (such as tyrosine and tryptophan) as well as charged groups. Their binding to GO primarily relies on two mechanisms: on one hand, π–π stacking interactions between the aromatic structures of the antibody and the GO surface enable stable adsorption. On the other hand, positively charged structures within the antibody (such as lysine side chains) can form electrostatic interactions with carboxyl groups on the GO surface, resulting in the formation of a non-covalent composite system. Such systems can enhance the stability and residence time of the drug in tumor-targeted delivery while maintaining the biological activity of the antibody [86].

Vaccine antigens, as protein-structured biomacromolecules, possess high polarity and charge characteristics, particularly due to exposed carboxyl groups, amino groups, and hydrophobic regions on their surfaces. In GO-based composite vaccine platforms, antigens can be electrostatically adsorbed onto the GO surface, while also participating in metal ion-induced (e.g., Zn^2+^) self-assembly to form nano-vaccine complexes. Such self-assembled systems enhance antigen stability, promote antigen presentation and dendritic cell uptake, and strengthen both humoral and cellular immune responses [87].

Nucleic acid molecules such as siRNA and plasmid DNA [88] possess a highly negative charge due to their phosphate backbone, which makes them readily adsorbed by positively charged materials under physiological conditions. After functionalization (e.g., grafting with polyethylenimine (PEI), or doping with nitrogen or metal elements), GO exhibits a positive surface charge and can form electrostatically adsorbed complexes with nucleic acids [89]. In addition, in certain systems, amino functional groups or doped materials provide covalent binding sites, allowing siRNA to be stably attached to the GO surface via covalent coupling or hydrogen bonding [90]. Incorporating aptamer structures can further impart tumor cell-specific recognition capability, making these platforms suitable for precision therapy and CRISPR/Cas9 delivery systems. Such GO-based carriers have been widely applied in RNA interference, gene editing, and apoptosis-inducing therapies.

In summary, to address the structural characteristics and biological delivery requirements of macromolecular drugs, researchers have developed a variety of GO-based drug-loading strategies. These primarily include:Non-covalent adsorption methods based on mixed adsorption;Ultrasonication-assisted dispersion techniques;Electrospinning encapsulation methods;And covalent grafting approaches aimed at enhancing binding stability.

Each of these strategies offers distinct advantages in practical applications. Their selection and optimization must be tailored according to factors such as drug molecular properties, delivery pathways, physiological environments, and therapeutic targets. By systematically optimizing the choice of methods based on drug characteristics, release scenarios, and targeting strategies, a solid foundation has been established for constructing functionalized and controllable macromolecular delivery systems.

### 2.3. Targeted Delivery of GFNs

To overcome the limitations of traditional nanomaterials in drug delivery—such as low efficiency, nonspecific distribution, and potential toxicity—researchers have developed various functionalized GO-based nanodelivery systems. By applying surface modification techniques (such as targeting molecule conjugation, introduction of stimuli-responsive structures, and multifunctional integration), different carriers can be attached to the GO surface. For example, functionalized GO has been integrated with aptamer-targeting modules to improve tumor selectivity and gene delivery efficacy [91]. In another case, folate-functionalized chitosan/reduced-GO/nickel oxide nanocomposites have shown precise tumor targeting and pH-responsive doxorubicin release, enabling selective cytotoxicity against cancer cells [92]. Another example involved protein corona-engineered GO nanosheets conjugated with iRGD peptides, enabling deep tumor penetration and targeted photothermal-triggered doxorubicin release [93]. Additionally, Chitosan-functionalized GO nanoparticles have also been developed to enable both fluorescence imaging and anti-inflammatory drug delivery, enabling real-time tracking of apoptotic processes [94]. Furthermore, stimuli-sensitive GO nanomaterials modified for endogenous (e.g., pH, ROS) or external (e.g., NIR light) triggers have demonstrated smart-release and extended circulation times [95]. These systems demonstrate notable advantages in enhancing drug targeting specificity, controlled-release efficiency, and imaging/monitoring capabilities, highlighting the important biomedical application potential of functionalized GO modifications.

#### 2.3.1. Enhanced Targeting via Ligand Surface Modification

By conjugating small-molecule ligands, peptides, aptamers, or antibodies as biorecognition units, the selective recognition ability of graphene carriers toward tumor cells can be significantly enhanced. For example, Jiahui Lu et al. [53] utilized GAD extracted from Ganoderma lucidum, loaded onto PEG-modified and EGFR antibody-functionalized GO (GO-PEG@GAD). This system significantly improved the tumor-specific accumulation of GAD and enhanced its therapeutic efficacy. Reyhan Yanikoglu et al. [72] developed a three-dimensional GO-based drug delivery platform by conjugating folic acid (FA) to GO nanosheets loaded with MTX, thereby enhancing targeting specificity toward breast cancer cells. Experimental results demonstrated that this system exhibited significantly higher cytotoxicity against tumor cells compared to free MTX. Furthermore, Chen et al. [96] reported a dual-targeting SERS-encoded GO nanocarrier functionalized with magnetic nanoparticles and anti-HER2 antibodies, enabling targeted co-delivery of doxorubicin and 9-aminoacridine with real-time SERS tracking, thereby achieving enhanced tumor selectivity and reduced side effects.

Li et al. [97] developed an NPF@DOX system by assembling PEGylated nano-GO with a fibroblast activation protein (FAP)–targeting peptide. Under combined photothermal therapy, this system demonstrated excellent antitumor efficacy. Similarly to this peptide-mediated approach, Chuang et al. designed a GRP-conjugated magnetic GO for dual-targeted delivery of irinotecan and SLP2 shRNA to glioblastoma, achieving efficient brain tumor accumulation, marked tumor growth inhibition, and prolonged survival in vivo [98]. Similarly, Nizamudin et al. conjugated Fe_3_O_4_ nanoparticles with an aptamer to form an MGO carrier, achieving specific recognition of MCF-7 cells, which may help improve therapeutic selectivity and provide a reference approach for future precision oncology treatments [99]. In a related strategy, Liu et al. developed an AS1411 aptamer-functionalized GO composite targeting nucleolin (C23), which significantly enhanced doxorubicin uptake and cytotoxicity in HeLa cells while sparing normal cells [100]. Complementing this, Wang et al. prepared a MUC-1 aptamer-modified hierarchical GO nanotheranostic agent enabling targeted doxorubicin delivery with switchable fluorescence imaging for precise breast cancer therapy [101]. Likewise, Kim et al. [102] fabricated an HB5 aptamer-tagged PEGylated GO nanocarrier for targeted doxorubicin delivery to HER2-positive breast cancer cells. The HB5 aptamer exhibited high binding affinity to HER2, promoting selective cellular uptake and markedly enhancing cytotoxicity, while PEG modification improved stability and biocompatibility.

Other studies, such as the DOX@NGOBBN system, have demonstrated targeted drug release for oral squamous cell carcinoma along with good stability, showing potential clinical value in enhancing therapeutic selectivity and reducing side effects for future oral cancer treatments [103]. Additionally, Li et al. [86] constructed a TRA/GO complex formed through non-covalent interactions, which enhanced the binding activity of anti-HER2 antibodies to breast cancer cells without showing significant toxicity. This provides a feasible strategy for targeted therapy of HER2-positive tumors. In parallel, Zhang et al. reported an anti-EpCAM antibody-functionalized GO for selective colorectal cancer targeting, achieving improved uptake and therapeutic efficacy [104]. In summary, surface functionalization techniques enable GFNs to achieve highly specific cell recognition and drug accumulation, thereby not only improving therapeutic selectivity and efficiency but also providing a solid material foundation for the development of precision delivery systems in personalized cancer therapy.

#### 2.3.2. Stimuli-Responsive Release: pH-Triggered Controlled Release Strategy

Tumor tissues typically exhibit a slightly acidic microenvironment (pH 5.0–6.5), providing a targetable condition for the development of pH-responsive nanodrug delivery systems.

Researchers have achieved controlled drug release in the acidic tumor microenvironment by introducing pH-sensitive linkages (such as hydrazone bonds, carboxyl groups, and amide bonds) and constructing stimuli-responsive encapsulation and dissociation systems. For instance, Dilip O et al. [47] developed a pH-responsive system composed of HA, GQDs, and adipic acid dihydrazide (ADH), which efficiently released PTX for breast cancer treatment. This was due to PAC protonation at low pH, reducing hydrophobic interactions with GQDs, and acid-induced weakening of ADH–HA linkages, facilitating targeted release. Similarly, Ma et al. [105] designed a DOX-hyd-PEG-FA/NGO system, combining FA targeting and acid-triggered release mechanisms, achieving highly efficient drug delivery. Here, acid-labile hydrazone bonds undergo rapid cleavage in the tumor’s acidic milieu, while DOX protonation concurrently weakens π–π stacking with NGO, thereby synergistically promoting site-specific release. Furthermore, Liu et al. [106] developed a DHA-GO-Tf system, utilizing Dihydroartemisinin (DHA) and transferrin (Tf) for dual functionalization. This system enables efficient drug release in the acidic lysosomal environment while simultaneously enhancing tumor cell uptake and cytotoxicity, this exploits Tf’s pH-dependent ferric binding, where endocytosis into acidic lysosomes triggers ferric release and reduction to ferrous ions, finally increasing reactive oxygen species (ROS)production and enhancing DHA cytotoxicity.

In addition, Qiu et al. [107] developed a DOX delivery system based on highly fluorescent GQDs, which achieved precise controlled release in the cancer cell microenvironment through pH-dependent drug–carrier interactions. In this design, the acidic tumor microenvironment markedly accelerates drug release by enhancing DOX solubility and protonating amino groups on both DOX and GQDs, thereby weakening hydrogen bonding and facilitating dissociation, while release remains minimal under physiological pH. Similarly, dual-responsive complexes (NCGO-FA) have also been shown to exhibit excellent controlled release properties under acidic conditions, owing to protonation-induced weakening of π–π stacking, disruption of electrostatic balance, and accelerated molecular diffusion at elevated temperatures [108]. Kitae Ryu et al. [109] modified PEI-rGO with a charge-convertible pH-responsive polymer (PKE), enabling the system to release DOX under lysosomal microenvironment conditions (pH 5.0, presence of glutathione). This system demonstrated excellent lysosome-targeting and pH-responsive release capabilities. This effect derives from pH-induced charge reversal that promotes tumor cell uptake, followed by lysosomal acidity and glutathione disrupting DOX–rGO interactions, enabling rapid intracellular release and potent antitumor activity. Similarly, Thangavelu Kavitha et al. [110] covalently attached the pH-sensitive polymer PDEA to the GO surface via in situ ATRP (atom transfer radical polymerization), successfully achieving efficient release of CPT in the acidic tumor microenvironment, this was attributed to protonation of CPT’s ring nitrogen at lower pH, which increased its hydrophilicity and solubility and weakened hydrophobic interactions with GO-PDEA, promoting release.

## 3. Applications of GFNs Delivery Systems in Different Diseases

### 3.1. Cancer Therapy

#### 3.1.1. Breast Cancer

Breast cancer is the most common malignant tumor among women worldwide and is one of the leading causes of cancer-related deaths in women [111]. According to statistics, in 2020, there were over 2.26 million newly diagnosed breast cancer cases globally, with approximately 685,000 deaths [112]. The incidence rate continues to rise, posing a serious threat to women’s health. Currently, breast cancer treatment primarily relies on a combination of approaches, including surgery, radiotherapy, chemotherapy, hormone therapy, and targeted therapy. However, traditional drug therapies face multiple challenges in clinical applications [113,114]. Firstly, most chemotherapeutic agents lack tumor specificity and exhibit nonspecific distribution throughout the body, easily causing damage to normal tissues and inducing adverse effects such as cardiotoxicity and hepatic and renal dysfunction. Secondly, the complex tumor microenvironment—characterized by factors such as low pH and high glutathione (GSH) levels—can accelerate drug degradation. This leads to the rapid clearance of active ingredients in the bloodstream, making it difficult to maintain an adequate drug concentration within tumor tissues and consequently reducing therapeutic efficacy.

To address the challenges mentioned above, researchers are actively exploring novel nanodrug delivery systems. GFNs can be functionalized through surface modification to enable both covalent and non-covalent binding with various drugs and targeting molecules [115,116]. This facilitates the construction of highly efficient, intelligent, and responsive targeted drug delivery systems tailored for cancer therapy [117].

Breast cancer tumor tissues typically exhibit a slightly acidic microenvironment, which provides a biological target for designing pH-sensitive nanodrug delivery systems.Aliyeh Ghamkhari developed a GO-PLA-PEG nanocarrier system, achieving efficient DOX loading through electrostatic and non-covalent interactions. This system enables rapid drug release under acidic conditions and induces apoptosis in 4T1 breast cancer cells, demonstrating promising therapeutic effects with reduced side effects [72]. The team led by Dilip O. Morani develop7ed a PAC@HA-ADH-GQDs system with a particle size of 25–50 nm and a drug-loading capacity as high as 93.56%. The system exhibited a release rate of up to 70% at pH 5, effectively targeting MCF-7 cells through CD44 receptor-mediated mechanisms, with significantly enhanced cellular uptake efficiency [105]. Chia-Jung et al. developed a GO-based nanoplatform functionalized with pH-sensitive materials, enabling the precise release of oligonucleotides through a competitive binding mechanism between rhodamine and GO under acidic conditions [118]. In addition, multiple studies have focused on the pH-sensitive release properties of commonly used chemotherapeutic agents such as DOX and PTX. For example, a GO-TD-Fe_3_O_4_@PEG system modified with triazine dendrimers achieved pH-responsive release of DOX and demonstrated good cellular uptake and pro-apoptotic effects in breast cancer cells (MCF-7) [119]. The GO-PAMAM system utilizes the pH-responsive properties of polyamidoamine (PAMAM) dendrimers to deliver QSR, exhibiting enhanced cytotoxicity in MDA-MB-231 breast cancer cells [50]. Armin Rahnama Rad introduced hydrazone linkages and FA ligands into the system, enhancing physiological stability through PEGylation while enabling precise drug release in the slightly acidic tumor microenvironment [120].

By modifying the GO surface with targeting ligands or aptamers, its selective uptake by breast cancer cells and therapeutic efficacy can be significantly enhanced. For example, Nizamudin Awel Hussien developed an aptamer-conjugated magnetic graphene oxide (MGO@APT) carrier that utilizes a MUC1 aptamer to recognize MCF-7 cells, achieving a high drug encapsulation efficiency of up to 95.75% and a high release rate under acidic conditions, further improving its targeted therapeutic effect [99]. Another study developed a GO-HA-Fe_3_O_4_ composite, co-loading DOX and PTX, and utilized a HA-mediated CD44 receptor targeting mechanism to enhance uptake by MDA-MB-231 cells. The system showed limited effects on CD44 low-expression cells (BT-474), indicating good targeting specificity [121]. Furthermore, a patent (AU2018269742B2) discloses a ligand-functionalized lipid-coated nanoparticle co-delivering an indoleamine 2,3-dioxygenase (IDO) inhibitor and an immunogenic cell death (ICD)-inducing agent (e.g., doxorubicin, oxaliplatin). This design enables targeted accumulation at orthotopic tumors, integrating immune modulation with chemotherapy to enhance efficacy and reduce off-target effects. Abutaleb Alinejad further proposed the design of an intelligent GO-PEI system through density functional theory (DFT) simulation. This system integrates pH-responsive and near-infrared radiation (NIR) light-triggered dual-controlled release mechanisms, providing both theoretical and experimental support for photo-responsive controlled release applications [122]. In the detection aspect, a clinical study (NCT07034248) by Chia-Hsun Hsieh et al. developed a nitrogen, sulfur-doped graphene quantum dot/3D gold nanoparticle (NSGQDs/AuNP) biosensor conjugated with Phaseolus vulgaris leucoagglutinin (PhaL) for label-free breast cancer cell detection, achieving a detection limit of 6 cells mL^−1^ within a 5–2500 cells mL^−1^ range, and demonstrating high sensitivity and stability with strong potential for early clinical diagnosis. We summarize in Table 1 the different GFNs complexes that were developed as nanocarriers for breast cancer therapy.

#### 3.1.2. Lung Cancer

Lung cancer is one of the most prevalent and deadliest malignant tumors worldwide, often diagnosed at advanced stages with a generally poor prognosis. Although targeted therapies and immunotherapies have improved treatment outcomes to some extent, traditional drug therapies still face significant challenges, including:Widespread drug distribution throughout the body,Strong side effects and toxicity,Poor tumor targeting,And high drug resistance.

Graphene-based carriers can be engineered with targeting ligands, stimuli-responsive elements, or synergistic therapeutic agents. These modifications improve drug accumulation in lung cancer tissues and support controlled-release delivery as well as multimodal therapeutic integration.

Wei et al. developed hollow GO microcapsules (GOMs) using a liquid nitrogen cavitation template-free strategy for the efficient delivery of PTX [43]. These microcapsules feature a hollow structure and high specific surface area, significantly improving drug-loading capacity. They also exhibit good water dispersibility and enable pH-responsive release in the acidic tumor microenvironment. The system demonstrated notable inhibitory effects on A549 lung cancer cells while exhibiting low toxicity toward normal cells. Zohreh Naseri et al. [123] prepared NGO/PLA–PEG composites by combining nano-GO with PLA–PEG diblock copolymers using a probe sonication method. The resulting composite exhibited uniform particle size and strong colloidal stability. This system was also applied for the sustained release of PTX. In vitro experiments confirmed its efficient uptake by A549 lung cancer cells and demonstrated notable anticancer activity.

To further enhance the targeting specificity and stimuli-responsive release efficiency in lung cancer treatment, Liu et al. [124] designed a redox-responsive nanodelivery system based on GO (NGO-SS-HA). In this platform, HA is grafted onto the GO surface via disulfide bond-containing linkers, enabling cleavable drug release under high glutathione (GSH) conditions. The system effectively delivers gefitinib. Additionally, HA provides CD44 receptor-mediated targeting capability, allowing efficient endocytosis in A549 lung cancer cells. Animal studies demonstrated that NGO-SS-HA exhibits significant tumor accumulation, strong tumor suppression effects, and minimal toxicity to normal tissues. We summarize in Table 2 the different GFNs complexes that were developed as nanocarriers for lung cancer therapy.

#### 3.1.3. Cervical Cancer

Cervical cancer is one of the most common malignant tumors of the female reproductive system worldwide, with particularly high incidence and mortality rates in developing countries. Although surgery, radiotherapy, and chemotherapy are the primary treatment options, drug therapy still faces numerous challenges, including:Widespread drug distribution in the body,Lack of tumor specificity,High potential for drug resistance,And significant side effects on normal tissues.

In recent years, synergistic systems combining multiple stimuli-responsive mechanisms and therapeutic modalities have become an important development direction for nanotherapy in cervical cancer. Li et al. [125] constructed a multifunctional graphene oxide nano drug carrier (GO@LM-SP-FA) that achieves tumor targeting through folic acid functionalization and integrates both pH and near-infrared (NIR) dual-responsive capabilities. In HeLa cervical cancer cells and MCF-7 breast cancer cells, GO@LM-SP-FA/DOX demonstrated significant antitumor activity, with photothermal therapy (PTT) further enhancing the therapeutic effect. Similarly, Peiwei Gong et al. developed ultrasmall fluorinated GO (FGO), which also possesses pH- and NIR-responsive properties and integrates a switchable fluorescence monitoring function. By combining this with photothermal enhancement effects, the system achieved highly effective cytotoxicity against HeLa cells, demonstrating excellent comprehensive therapeutic performance [126].

In addition to multi-responsive systems, surface functionalization to enhance tumor cell targeting and drug delivery efficiency is also a key research focus. Lu et al. [53] developed a targeting antitumor nanocomposite (GO-PEG@GAD) featuring dual modification with PEG and an EGFR aptamer, enabling active targeting recognition of HeLa cervical cancer cells. This system not only demonstrated a high drug-loading capacity (77.3%) and encapsulation efficiency (89.1%) but also achieved sustained release of GAD for over 100 h, exhibiting excellent controlled-release capability and anticancer efficacy. Wu et al. [127] developed a GO–HA nanoplatform that utilizes HA for CD44 receptor-mediated targeting, effectively achieving selective delivery of DOX to HeLa cells. This approach enhances both the therapeutic safety and precision of the system. We summarize in Table 3 the different GFNs complexes that were developed as nanocarriers for cervical cancer therapy.

#### 3.1.4. Liver Cancer

Liver cancer is a malignant tumor that arises from hepatic parenchymal cells. It primarily includes hepatocellular carcinoma, as well as cholangiocarcinoma and other subtypes. It is associated with chronic liver injury factors such as hepatitis B or C virus infection, long-term alcohol abuse, or metabolic syndrome. Due to its insidious progression and non-specific early symptoms, liver cancer is often diagnosed at intermediate or advanced stages, resulting in poor prognosis. It ranks as one of the most lethal cancers worldwide.

Smart responsive systems that integrate external physical stimuli with endogenous tumor microenvironment characteristics are becoming an important strategy to enhance liver cancer treatment efficiency. Raquel O. Rodrigues et al. [128] designed a multifunctional magnetic nanoplatform based on graphene-coated magnetic core–shell structures. This system exhibits high magnetic saturation intensity, excellent drug-loading capacity, and enables efficient hyperthermia under an alternating magnetic field. It also achieves DOX release in the acidic tumor microenvironment, demonstrating strong cytotoxicity against liver cancer cells while maintaining low side effects. Zhao et al. [129] developed the GON/CS/CS-DMMA system, which employs a charge-reversible strategy. This system can switch to a positive charge in the slightly acidic tumor microenvironment (pH 6.5–6.9), promoting cellular uptake. Under even more acidic conditions, it removes the chitosan coating, accelerating DOX release. The system demonstrates high pH responsiveness and strong anticancer efficacy. Yuling He et al. further proposed a GO-based pH-responsive, charge-reversible multilayered carrier designed for the co-delivery of DOX and shRNA targeting the ABCG2 gene [130]. This system not only accelerates the release of chemotherapeutic drugs under acidic conditions but also effectively downregulates the expression of drug-resistance genes, thereby significantly enhancing liver cancer treatment sensitivity and helping to overcome drug resistance.

Enhancing targeting specificity and physiological stability through chemical modification is a key strategy for enabling the biomedical application of graphene-based drug delivery systems. Pan et al. achieved this by modifying the GO surface with carboxymethyl chitosan (CMC) and lactobionic acid (LA), endowing the system with selective recognition capability toward liver cancer cells [131]. This system accelerates DOX release under acidic conditions while maintaining sustained release in normal tissue environments, thereby enhancing anticancer efficacy and reducing side effects. Similarly, Erqun Song et al. [132] developed an HA-GO carrier that combines CD44 receptor-mediated active targeting with acid-responsive release properties. This system not only promotes DOX accumulation and release in liver cancer cells but also exhibits good biocompatibility with normal cells. Zhang et al. [133] modified GO with heparin to enhance its hemocompatibility and drug stability. The system enables rapid DOX release under mildly acidic conditions while maintaining sustained release in neutral environments. Additionally, it helps reduce common cardiopulmonary toxicity associated with chemotherapy and demonstrates dual cytotoxic effects against both liver cancer and breast cancer cells. We summarize in Table 4 the different GFNs complexes that were developed as nanocarriers for liver cancer therapy.

#### 3.1.5. Other Cancers

Li et al. [103] loaded the targeting peptide BBN-AF750 and DOX onto the surface of GO, creating a pH-responsive nanosystem capable of accelerated drug release under acidic conditions. Results showed that this carrier can specifically recognize oral squamous carcinoma cells and efficiently release DOX intracellularly, thereby enhancing tumor cell cytotoxicity while maintaining normal cell safety. Additionally, Li et al. [97] developed a pH-responsive nano-GO carrier targeting FAP, named NPF@DOX, for the treatment of oral squamous cell carcinoma (OSCC). This system utilizes PEG modification to improve stability and is loaded with the chemotherapeutic drug DOX. Under NIR laser irradiation, it exhibits high photothermal conversion efficiency (up to 52.48%). The carrier demonstrates specific tumor tissue accumulation and accelerated drug release under acidic and photothermal conditions, achieving a synergistic chemotherapeutic and photothermal anticancer effect. It significantly inhibits tumor growth while showing low toxicity to normal tissues.

Forough Alemi et al. [134] developed a pH-responsive drug delivery system based on functionalized GO (FGO) for the treatment of osteosarcoma. By modifying the GO surface and loading it with DOX, the system achieved faster drug release under the acidic conditions of the tumor microenvironment, effectively enhancing both DOX cellular uptake and apoptosis induction. In vitro experiments demonstrated that FGO-DOX exhibited stronger cytotoxic effects compared to free DOX, while maintaining good biocompatibility. Additionally, Linlin Dong et al. [135] designed magnetic GO microcapsules (MGOMCs) that integrate flexible structures, Fe_3_O_4_ magnetic targeting, and a disulfide bond redox-responsive mechanism. These microcapsules enabled dual-triggered controlled release in HeLa cells, improving the spatiotemporal precision of drug delivery. We summarize in Table 5 the different GFNs complexes that were developed as nanocarriers for other cancers’ therapy.

### 3.2. Treatment of Neurological Disorders

#### 3.2.1. Alzheimer’s Disease

Alzheimer’s disease (AD) is a neurodegenerative disorder characterized by β-amyloid (Aβ) deposition and abnormal tau protein phosphorylation. It involves a complex pathogenesis and currently has limited therapeutic options.In recent years, GFNs have shown great potential in mechanistic intervention, brain-targeted therapy, and biological detection related to AD.

Oxidative stress and neuronal apoptosis triggered by Aβ deposition are among the key pathological mechanisms in the early stages of AD, while traditional antioxidant drugs show limited efficacy in brain tissue. To address this issue, Wang et al. developed a GO-based nanoformulation loaded with dauricine (Dau), which significantly inhibited Aβ1-42-induced oxidative damage in both in vitro and in vivo models, demonstrating strong neuroprotective effects [136].

The difficulty of drug penetration across the blood–brain barrier (BBB) is a major obstacle limiting the therapeutic efficacy of AD treatments. To enhance drug delivery efficiency to the brain, Fariba Mohebichamkhorami and Andrea Gabriele et al. [137] prepared an ultrasmall chitosan/graphene quantum dot (CS/GQD) nanoplatform using microfluidic technology. This system demonstrated good cellular permeability and brain-targeting capability, effectively improving cognitive function in AD model rats. Similarly, plant-derived ctGQDs synthesized through green synthesis using Clitoria ternatea extract also exhibit the ability to cross the BBB. In animal models, ctGQDs effectively improved cognitive impairment, expanding the application potential of natural graphene-based materials in AD treatment [138].

Current AD diagnostic methods, such as ELISA, face challenges like complex procedures and limited sensitivity, making them insufficient for early screening and high-throughput detection needs. To improve detection efficiency, Huang et al. [139] developed a fluorescence quenching immunosensor platform based on GO and anti-tau antibodies. This system enables highly sensitive quantitative detection of tau protein, with a detection limit as low as 0.14 pmol/mL. It also eliminates the need for enzyme-labeled antibodies, offering the advantage of a simpler and faster workflow. Benítez-Martínez et al. [140] developed a fluorescence sensing platform based on nitrogen-doped graphene quantum dots (N-GQDs) and acetylcholinesterase (AChE). This system enables high-sensitivity detection of AChE activity and its inhibitor Tacrine, with a detection limit of 1.22 μM. It is well-suited for anti-AD drug screening and therapeutic efficacy evaluation. Similarly, Srishti Sharma et al. [141] developed a fluorescence sensing platform based on NGQDs for sensitive monitoring of the inhibitory effects of novel anti-AD drugs on AChE. Among the tested candidates, PC-37 demonstrated the strongest inhibitory activity, further validating the practicality and effectiveness of such sensing platforms in anti-AD drug screening and evaluation. 

#### 3.2.2. Parkinson’s Disease (PD)

PD is characterized by the degenerative loss of central dopaminergic neurons, which severely impacts patients’ motor function. Due to the presence of the BBB, most drugs cannot effectively reach the brain, making this a major obstacle that limits therapeutic efficacy. To address this challenge, Xiong et al. [142] developed a lactoferrin (Lf)-functionalized GO nanosystem (Lf-GO-Pue), successfully delivering puerarin (Pue) to the brain. The system utilizes Lf receptor-mediated trans-BBB transport for precise targeting, significantly improving motor dysfunction symptoms in PD model rats. This demonstrates its potential in targeted drug delivery for PD treatment.

Structural damage to neural cells and insufficient regenerative capacity are also key issues in PD and other neurological disorders. To address this, Ankor González-Mayorga et al. [143] investigated the application of rGO microfibers as neural growth scaffolds. Their findings showed that these microfibers possess good electrical conductivity and structural stability, functioning as a physical guidance platform for nerve regeneration. They effectively promote the repair of damaged spinal cord tissue, highlighting the potential of graphene materials in neural tissue engineering. We summarize in Table 6 the different GFNs complexes that were developed as nanocarriers for PD and AD therapy.

### 3.3. Cardiovascular Diseases

Cardiovascular diseases are among the leading causes of mortality worldwide, encompassing various pathological processes such as myocardial infarction, atherosclerosis, and vascular regeneration disorders. Current treatment approaches still show clear limitations in aspects such as tissue repair capacity, controlled drug release, targeting specificity, and vascular function regulation. In recent years, GFNs have demonstrated unique advantages in cardiovascular disease treatment due to their excellent electrical conductivity, mechanical strength, and surface functionalization potential. To overcome challenges such as inefficient tissue repair, poor drug targeting, and uncontrolled release, researchers have developed multifunctional therapeutic platforms using the unique properties of graphene. These systems combine capabilities including structural support, smart controlled release, and anti-inflammatory tissue repair, offering comprehensive solutions for cardiovascular therapy.

In the field of tissue regeneration and myocardial repair, graphene’s mechanical strength and electrical conductivity provide an ideal foundation for constructing engineered materials. Xiong et al. [144] developed a conductive, elastic, and vascularizable composite patch by incorporating coronary artery structures into a porous GO/polypyrrole hydrogel. This system enables pre-vascularized reconstruction and electrical integration within myocardial infarction regions, significantly improving myocardial functional recovery. Li et al. [145] constructed an injectable self-healing hydrogel using silk fibroin and GO, combined with cardiac progenitor cells (CPCs) and growth factors for synergistic therapy. This system offers good cell adhesion and oxidative stress protection, significantly enhancing cardiomyocyte survival rates and myocardial marker gene expression. These graphene-based scaffold and hydrogel construction strategies provide a stable, conductive, and injectable microenvironment that supports cardiac regeneration.

On the other hand, graphene platforms have also demonstrated strong capabilities in drug delivery and localized regulation. Sandeep Kumar Yadav et al. developed a GO-Gel-ATR nanosystem that enables smart controlled release of atorvastatin in atherosclerotic plaque regions (pH 6.8), enhancing drug efficacy control and plaque stability [146]. The team led by Duygu Kaya designed an rGO-enhanced conductive membrane that not only provides stable release of irbesartan but also improves membrane conductivity and biocompatibility, offering a potential solution for implantable cardiovascular devices [147]. The Tabish team developed two types of NO release systems: SNO-Cys@PGO and PEI-PEG@GO. These systems achieve stable and safe NO delivery through thiol group chemical reactions and endogenous precursor catalysis, respectively. Both platforms effectively promote endothelial cell proliferation and inhibit abnormal smooth muscle cell proliferation, simulating a functional microenvironment of healthy blood vessels. These systems show promising potential for applications in vascular regeneration and stent repair [148,149]. In addition to these preclinical advances, clinical translation of graphene-based systems for cardiovascular disease diagnosis is also underway. A prospective randomized controlled trial (NCT04390490) in China is designed to evaluate the sensitivity, precision, and effectiveness of a photoelectrochemical immunosensor composed of graphene quantum dots combined with Si nanowires for the early diagnosis of acute myocardial infarction. Using a parallel-group, quadruple-blinded design, the study will recruit adult patients presenting with chest pain to assess this device against established diagnostic approaches, representing an important step toward clinical implementation of graphene-based diagnostic platforms in cardiovascular medicine. We summarize in Table 7 the different GFNs complexes that were developed as nanocarriers for cardiovascular diseases therapy.

### 3.4. Bacterial Infections and Inflammation

Bacterial infections are widespread in chronic wounds, surgical implants, and immune-related complications. Especially with the rise of drug-resistant strains, traditional antimicrobial therapies face challenges such as declining efficacy and difficulty maintaining effective local drug concentrations. To enhance antibacterial efficiency and control drug release, researchers have developed multifunctional antibacterial nanoplatforms based on GFNs. These platforms have achieved notable progress in areas including infection control, sustained antibiotic release, and promotion of tissue repair.

#### 3.4.1. Bacterial Infections

In practical applications, local wound dressings often face limitations such as short drug efficacy duration, poor moisture retention, and insufficient mechanical properties, which affect treatment outcomes. To address these challenges, Negar Hosseini Darabi et al. [150] proposed an electrospun nanofiber membrane composed of GO and silver nanowires for wound dressings. This system offers good mechanical strength and moisture retention, while also achieving sustained release of CIP. It significantly promotes both antibacterial activity and wound healing. Yang et al. [67] developed a GO/chitosan/polyvinyl alcohol (CS/PVA) electrospun membrane capable of controlled delivery of ciprofloxacin (CIP) and its hydrochloride form (CipHCl). By regulating the initial burst release phenomenon, the system enhances drug release sustainability and improves antibacterial efficacy.

Traditional antibiotics often suffer from large fluctuations in blood drug concentration and short duration of action, limiting their effectiveness in sustained infection control. To extend drug efficacy and enhance loading capacity, Himanshu Pandey et al. [64] prepared a methanol derived graphene (MDG) material capable of efficiently loading gentamicin sulfate (2.57 mg/mg). The system enables pH-responsive release and demonstrated synergistically enhanced antibacterial effects against Escherichia coli. Proma Bhattacharya et al. further confirmed that the rGO/CIP composite system exhibits significant synergistic antibacterial activity against Pseudomonas aeruginosa, demonstrating that graphene can serve as an effective antibiotic-enhancing carrier [151].

With repeated use and in complex infection environments, traditional antibacterial materials are prone to increased bacterial resistance and declining antibacterial activity, limiting their potential for implantable material applications. To address this, Wu et al. [65] developed a magnetic GO (MGO) platform modified with Fe_3_O_4_ nanoparticles. By grafting gentamicin sulfate onto the MGO surface via amide coupling reactions, they constructed a GS-MGO system featuring both magnetic responsiveness and sustained antibacterial performance. This material demonstrated significant antibacterial effects against various Gram-positive and Gram-negative bacteria and maintained high efficacy even after multiple cycles of use. Ling et al. [71] constructed a co-modified coating of GO and DEX-loaded liposomes on the surface of polyetheretherketone (PEEK) material. This coating not only promoted adhesion, proliferation, and differentiation of osteoblasts (MC3T3 cells) but also provided excellent antibacterial properties. The system is suitable for surface modification of implantable medical devices with a high risk of infection. In addition to these material-based antibacterial strategies, clinical research has also been initiated on graphene-based diagnostic platforms. A prospective diagnostic trial (ChiCTR2500097286) sponsored by the First Affiliated Hospital of Army Medical University aims to evaluate a THz metamaterial–graphene composite chip for bacterial liquid-phase detection and real-time inactivation in patients with bacteremia or urinary tract infections, as well as in healthy volunteers. Using blood culture as the gold standard, the study will assess the chip’s ability to perform qualitative and quantitative bacterial analysis, with recruitment planned in China. Furthermore, patented technologies have also been developed. For example, a recent patent (US11981571) describes a graphene-based medical composite with broad-spectrum antibacterial activity and controlled-release capability, enabling long-lasting efficacy, improved stability, and reduced cytotoxicity. The material is applicable to wound dressings, implant coatings, and catheter surfaces, supporting long-term infection prevention in clinical settings. We summarize in Table 8 the different GFNs complexes that were developed as nanocarriers for bacterial infections therapy.

#### 3.4.2. Inflammation

Traditional anti-inflammatory drugs often face challenges such as poor water solubility, rapid metabolism in the body, and strong burst release effects when treating inflammation-related diseases. These factors result in short drug efficacy duration and a limited therapeutic window. To improve drug bioavailability and delivery stability, researchers have explored the use of GO-based platforms to develop novel anti-inflammatory drug delivery systems. Mohammad Saiful Islam et al. [70] proposed combining DEX with GO, which significantly improves its water solubility and allows for controlled release behavior. In in vitro models, the composite demonstrated a higher dissolution rate and enhanced anti-inflammatory activity. To further optimize carrier structure, Sun et al. [152] developed a controlled-release system for dexamethasone phosphomonoester (DEX-P) based on GO-modified chitosan (GO–CS). The drug was stably loaded via covalent bonding through phosphate ester linkages. This platform demonstrated steady release across different pH environments, significantly reducing burst release. It also showed good biocompatibility and low toxicity in bone marrow mesenchymal stem cells, providing a more durable and safer delivery solution for chronic inflammation control.

In addition to traditional anti-inflammatory drugs, small-molecule antibody delivery has attracted widespread attention in the treatment of autoimmune diseases and immune modulation therapies. However, such antibodies face challenges including instability in vivo, easy degradation, and uncontrolled release. To address these issues, Ni et al. [153] developed a sustained-release delivery system for anti-IL-10 receptor antibody (anti-IL10R) based on GO. This system leverages GO’s surface adsorption properties and pH-sensitive release capability, enabling precise antibody release under mildly acidic conditions while maintaining its immunobiological activity. We summarize in Table 9 the different GFNs complexes that were developed as nanocarriers for bacterial inflammation therapy.

### 3.5. Diabetes

Insulin is the core macromolecular drug for diabetes treatment; however, during in vivo delivery, it is highly susceptible to factors such as temperature, pH, and enzymatic degradation, leading to misfolding, aggregation, or inactivation. These issues significantly impact its bioavailability and therapeutic efficacy. To address this, researchers have extensively explored the potential of GFNs as insulin carriers. For example, GO offers excellent specific surface area, good surface modification capability, and environmental responsiveness. By introducing functional groups such as carboxyl, PEG, or PEI, it is possible to regulate surface charge, thereby inhibiting protein fibrillation and enhancing insulin stability and activity in vivo.

To achieve smart insulin release and adaptive control within the biological environment, Florina Teodorescu et al. [154] developed an rGO-doped poly(ethylene glycol) dimethacrylate (PEGDMA) hydrogel system. This system is capable of efficient insulin loading and enables on-demand release. The team led by Hakim Belkhalfa applied electrophoretic deposition technology to deposit an rGO/Ni(OH)_2_/insulin composite onto gold electrode surfaces. This approach further improved drug-loading efficiency and enabled release control via external electrical stimulation, offering a new pathway for multimodal insulin management [84].

Although injection remains the primary method for insulin therapy, it is highly invasive and often results in poor patient compliance. Therefore, developing oral delivery systems has become a key focus in diabetes treatment research. However, insulin is highly susceptible to inactivation in the gastrointestinal tract due to factors such as acidic conditions and enzymatic degradation. To overcome this challenge, Shabana Gul Baloch et al. developed an insulin-intercalated graphene oxide nanogel composite (In@GO NgC) via modified Hummers’ synthesis of GO, insulin intercalation, and emulsion polymerization with hydroxyethyl methacrylate/(N-iso-propylacrylamide, achieving high loading efficiency (90%), pH-dependent controlled release, and enhanced enzymatic stability in vitro [83]. Huang et al. [155] developed an ultrasound-synthesized GO microparticle system that exhibits high insulin-loading capacity under acidic conditions and shows pronounced release responsiveness under alkaline conditions. This feature further enhances the environmental adaptability of oral insulin delivery systems.

In terms of pharmacodynamic monitoring and feedback control, most existing delivery systems lack real-time response mechanisms, making it difficult to dynamically coordinate with blood glucose levels. To address this issue, Zhang et al. designed a GO–gold nanoparticle composite (AuNPs/GO) with optical responsiveness. This system can respond in real time to insulin concentration changes, showing potential for monitoring blood glucose regulation processes. It lays the groundwork for developing integrated “release–sensing–feedback” smart platforms [80]. Soganci et al. designed an rGO-patterned glucose sensor capable of real-time monitoring of glucose concentration changes. This platform features ultrafast response time and high sensitivity, laying a solid foundation for developing integrated “release–sensing–feedback” smart glucose regulation systems [156]. The same research team also developed a glucose sensor based on a CuS/rGO/GOx/GCE electrode, which can likewise monitor glucose concentration changes in real time, demonstrating strong potential for blood glucose regulation monitoring [157].

GFNs demonstrate multidimensional advantages in the delivery of macromolecular drugs such as insulin, especially in terms of maintaining drug stability, enhancing drug-loading capacity, and stimuli-responsive release—surpassing traditional carriers in these aspects. Through surface functionalization (e.g., PEG, PEI, carboxyl groups), GO can effectively prevent insulin aggregation and enzymatic inactivation, improving its in vivo activity and therapeutic efficacy. Additionally, functionalized hydrogel platforms and electrically controlled release systems enable on-demand regulation, enhancing both treatment flexibility and personalization potential. In oral delivery, GO-based nanogels, composite particles, and AuNPs–GO platforms have successfully overcome challenges related to gastrointestinal acidity and enzymatic degradation. These systems have significantly improved insulin bioavailability and release controllability, while also enabling monitoring of the glucose regulation response and the drug delivery process.

However, these strategies still face critical challenges in clinical translation. The long-term biocompatibility, degradability, and immunogenicity of GFNs require systematic evaluation. Their efficacy and safety in large animal models and human trials remain largely unverified. Additionally, for oral delivery systems, whether they can maintain sustained blood drug concentrations after crossing gastrointestinal barriers still needs further optimization. We summarize in Table 10 the different GFNs complexes that were developed as nanocarriers for diabetes therapy.

### 3.6. Drug Delivery for Imaging and Diagnostic Functions

Beyond its broad application in drug delivery and therapy, GFNs also demonstrate unique advantages in medical imaging. Due to their exceptional optical, electrical, and thermal properties, functionalized GFNs can serve as multimodal imaging probes for fluorescence, photoacoustic, and MRI techniques. Its strong NIR absorption capability makes it an effective contrast agent in photoacoustic imaging. Furthermore, graphene’s two-dimensional structure provides abundant surface modification sites, allowing it to be combined with various imaging functional molecules to create precise and biocompatible imaging systems. By integrating drug delivery with imaging and diagnostic functions, real-time monitoring and feedback control of the therapeutic process become achievable. For example, Li et al. anchored the GRPR-specific peptide AF750-6Ahx-Sta-BBN onto the surface of NGO via π–π stacking and hydrogen bonding, constructing a near-infrared imaging probe (NGO-BBN-AF750) targeting OSCC. In HSC-3 cell models, this system showed excellent targeted internalization and imaging performance [158]. Seon-Yeong Kwak et al. [159] developed a GO–peptide–QXL composite by linking GO with a fluorescence resonance energy transfer (FRET) quencher QXL570 and peptide chains, constructing an optical biosensing platform based on GO fluorescence for rapid and sensitive protease activity detection. In this system, the quencher is released upon protease action, restoring GO fluorescence and achieving an “off–on” detection mechanism. The platform demonstrated high response sensitivity to both trypsin and matrix metalloproteinase MMP-2, and was successfully applied to detect MMP-2 secreted by HepG2 cells.

## 4. Preclinical Research and Clinical Translation Challenges

GFNs, owing to their unique physicochemical properties, exhibit broad application prospects in biomedicine, especially in areas such as drug delivery, cancer diagnosis and therapy, and biological imaging. However, their potential biocompatibility issues and long-term in vivo safety remain key challenges restricting clinical translation. Although in vitro experiments and short-term in vivo studies suggest that surface functionalization (e.g., PEGylation, targeting molecule conjugation) can significantly improve biocompatibility, factors such as: Long-term metabolism in complex physiological environments, Types and densities of functional groups and Dosage and accumulation effects may all affect how GFNs interact with biological systems. Therefore, establishing clear long-term biosafety profiles and a standardized evaluation system is a critical prerequisite for advancing GFNs from the laboratory to clinical application.

### 4.1. Biocompatibility Assessment

The biocompatibility of GFNs is a critical aspect that must be clearly defined for their biomedical applications. Differences in size, surface chemistry, administration route, and functionalization lead to significant variation in cytotoxicity, immune response, and metabolic distribution. A deeper understanding of how these factors influence biocompatibility is essential for guiding GFn design and safety evaluation, promoting their rational application within biological systems. The in vivo biological behavior of GFNs is highly dependent on factors such as: surface structure, oxidation level and functionalization methods. These parameters not only affect cell membrane recognition and uptake patterns but also directly determine GFNs’ ability to regulate cell fate. The surface topological structure of GFNs plays a key role in modulating cell adhesion and alignment behavior, making it a central consideration in biocompatibility evaluation and material design.

Rowoon Park et al. [160] constructed nano-wrinkled GO films on elastic substrates and found that when the wrinkle amplitude reached a specific scale, the cytoskeleton of cells realigned along the wrinkle direction. This effectively induced cell adhesion and ordered extension. This phenomenon holds important implications for guiding cell alignment and organized tissue growth in tissue engineering.

The oxidation degree of GO determines its hydrophilicity and bioreactivity, thereby influencing both cellular uptake pathways and toxicity mechanisms. Nivedita Chatterjee et al. [161] compared GO samples with varying oxygen-to-carbon (O/C) ratios. They found that hydrophilic GO enters cells primarily through endocytosis, while more hydrophobic GO induces mitochondrial ROS generation and activates DNA damage-related pathways, such as p53 and MAPK. These effects lead to oxidative stress and cell apoptosis. This suggests that controlling oxidation degree is a key factor in optimizing GO biocompatibility.

In addition to its intrinsic structure, surface functionalization modification of GO also plays an important role in regulating its biological effects. María-Concepción Matesanz et al. compared the biocompatibility performance of GO modified with different structures of PEG (linear and 6-arm PEG) in various cell types [162]. The study showed that 6-arm PEG-GO could significantly induce ROS generation and cell cycle arrest in macrophages, while promoting cell adhesion and maintaining viability in osteoblasts, indicating that the structure of functional modifications can trigger cell type-specific biological responses.

In addition, the flexibility and conductivity of GFNs have also been utilized in designing nerve regeneration scaffolds. rGO microfiber materials, due to their electrical activity and good flexibility, have been applied in central nervous system injury models. Animal experiments show that they can guide axonal growth along the microfibers at spinal cord injury sites, constructing a biomimetic nerve repair platform, and are expected to serve as implantable tissue engineering support materials [143].

The distribution, metabolism, and clearance pathways of GFNs in vivo are directly related to their systemic toxicity and long-term biosafety. Zhang et al., [163] through radioactive isotope labeling, tracked the in vivo behavior of GO and found that after intravenous injection, GO primarily accumulates in the lungs and is gradually cleared through hepatic metabolism. Further co-incubation experiments with red blood cells showed that GO, at appropriate doses, had no significant impact on red blood cell membrane structure, suggesting it possesses a certain degree of blood compatibility.

### 4.2. Toxicity and Safety

Carbon-based nanomaterials—such as graphene, carbon nanotubes (CNTs), fullerene (C60), and GO—hold great promise for medical applications. These include cancer therapy, imaging diagnostics, and drug delivery, owing to their unique physicochemical properties. However, related studies have also shown that carbon-based nanomaterials may induce biological adverse effects such as cytotoxicity, organ accumulation, thrombosis, and immune interference, which severely limit their clinical translation. Therefore, systematically evaluating their dosage thresholds, physiological exposure mechanisms, and controllable strategies is essential to achieving safe application.

Studies have shown that the toxicity of GO exhibits clear dose dependency and varies with administration routes. Liu et al. [164] demonstrated in a murine model that when the intravenous dose of graphene oxide (GO) exceeds a critical threshold, it induces significant pulmonary inflammatory cell infiltration and oxidative stress, reflecting systemic toxicity. In their study, male CD-1 mice received GO as a single high dose of 2.1 mg kg^−1^, a single low dose of 0.3 mg kg^−1^, or multiple low doses of 0.3 mg kg^−1^ every other day for seven injections. A previous report from the same group showed that a single dose of 10 mg kg^−1^ was lethal within two weeks, highlighting the dose-dependent nature of GO toxicity. These included macrophage nodule formation and lymphocyte infiltration in pulmonary tissue, renal inflammatory lesions, hepatic inflammatory infiltration, oxidative stress–mediated cellular injury, and, in severe cases, mortality under repeated exposure. Yang et al. [165] further indicated that PEG-modified GO, when administered orally, can be completely excreted within 48 h with almost no organ accumulation; in contrast, intraperitoneally injected GO (200 nm) showed in vivo retention and induced local inflammation due to differences in surface charge and aqueous stability. This demonstrates that administration route and surface modification have a significant impact on GO’s toxicity. In Table 11, we present the toxicological profiles of GFNs via different administration routes.

At the cellular level, GO with different functional group modifications exhibits different toxicity. Ding et al. [170] found that pristine GO (p-GO) and carboxylated GO (GO-COOH) are relatively safe at low concentrations but show significant cytotoxicity at high concentrations, while polyethylenimine-modified GO (GO-PEI) shows strong toxicity even at extremely low concentrations. Mechanistic studies showed that p-GO can induce ROS-dependent cell apoptosis by inhibiting receptor-ligand binding, GO-COOH exerts toxicity through non-ROS mechanisms, and GO-PEI mainly causes cell death through membrane disruption. In addition, the effects of GO on the hematological system cannot be ignored. Studies have shown that even at a concentration of only 2 μg/mL, GO can significantly induce platelet aggregation and red blood cell hemolysis, increasing the risk of thrombosis [171]. In Table 12, we summarize the toxicological profiles of various functionalized graphene oxide (GO) variants.

To reduce the toxicity of GO during in vivo use, researchers have proposed various regulatory pathways and material optimization strategies. In terms of blood system safety, Sunil K. Singh et al. developed amino-modified GO (G-NH_2_), which did not induce thrombosis or red blood cell destruction in intravenous injection models, demonstrating good hemocompatibility [173]. Dola Sundeep et al. [172] proposed a green synthesis route by using ascorbic acid, galactose, and bovine serum albumin as reducing agents, instead of toxic chemicals like hydrazine. This method produced RGO trimers with high solubility and low cytotoxicity, offering a safer approach to GO preparation.

Soumen Das et al. [174] compared the cytotoxicity of GO and rGO in terms of their physicochemical properties. They found that GO’s high density of oxygen-containing groups leads to ROS accumulation and DNA damage. In contrast, moderate reduction can significantly lower toxicity while preserving solubility. In addition, studies on protein interactions also showed that GO-COOH causes the least structural disturbance to human serum albumin, while GO-PEI and p-GO significantly interfere with protein function. These results suggest that by regulating the degree of reduction, types of surface functional groups, and particle size distribution, the biosafety of GO-based materials can be effectively improved, laying a foundation for their application in clinical drug delivery and implantable therapies.

## 5. Conclusions and Outlook

In recent years, research on graphene-based composites for drug delivery has expanded significantly. It now covers various aspects such as material design, surface modification, drug-loading mechanisms, and targeted therapy. This paper systematically reviews the customized application progress of graphene and its derivatives in the treatment of various diseases, including cancer, neurological disorders, cardiovascular diseases, infection and inflammation regulation, as well as diabetes. Strategies such as ligand modification, aptamer conjugation, magnetic integration, and pH-responsive design have greatly improved the drug-loading efficiency and release precision of graphene-based carriers. These approaches have also enhanced cellular targeting and therapeutic outcomes. At the same time, this paper summarizes the in vivo distribution, metabolic behavior, and potential toxicity mechanisms of graphene materials, emphasizing the critical influence of factors such as dosage, oxidation degree, functional group types, and administration routes on their biocompatibility.

Although a variety of functional modification strategies for graphene have been established—such as PEGylation, peptide modification, antibody conjugation, and green reduction—which have significantly improved the material’s dispersibility, targeting ability, and biocompatibility, there are still certain limitations in customized design. On one hand, most material modifications focus on enhancing a single property, lacking integrated cross-functional construction. On the other hand, there is still no unified evaluation system for the stability and synergy of different modification methods in actual physiological environments. Future development should focus on coupling material structure, biological function, and in vivo response. This will support the creation of smart drug delivery platforms with controlled degradability, tissue-specific targeting, and multifunctional therapeutic capabilities. At the same time, systematic safety validation in large animal models and preclinical systems should be accelerated to advance the practical clinical translation of these materials in precision medicine applications.

## Figures and Tables

**Figure 1 pharmaceuticals-18-01245-f001:**
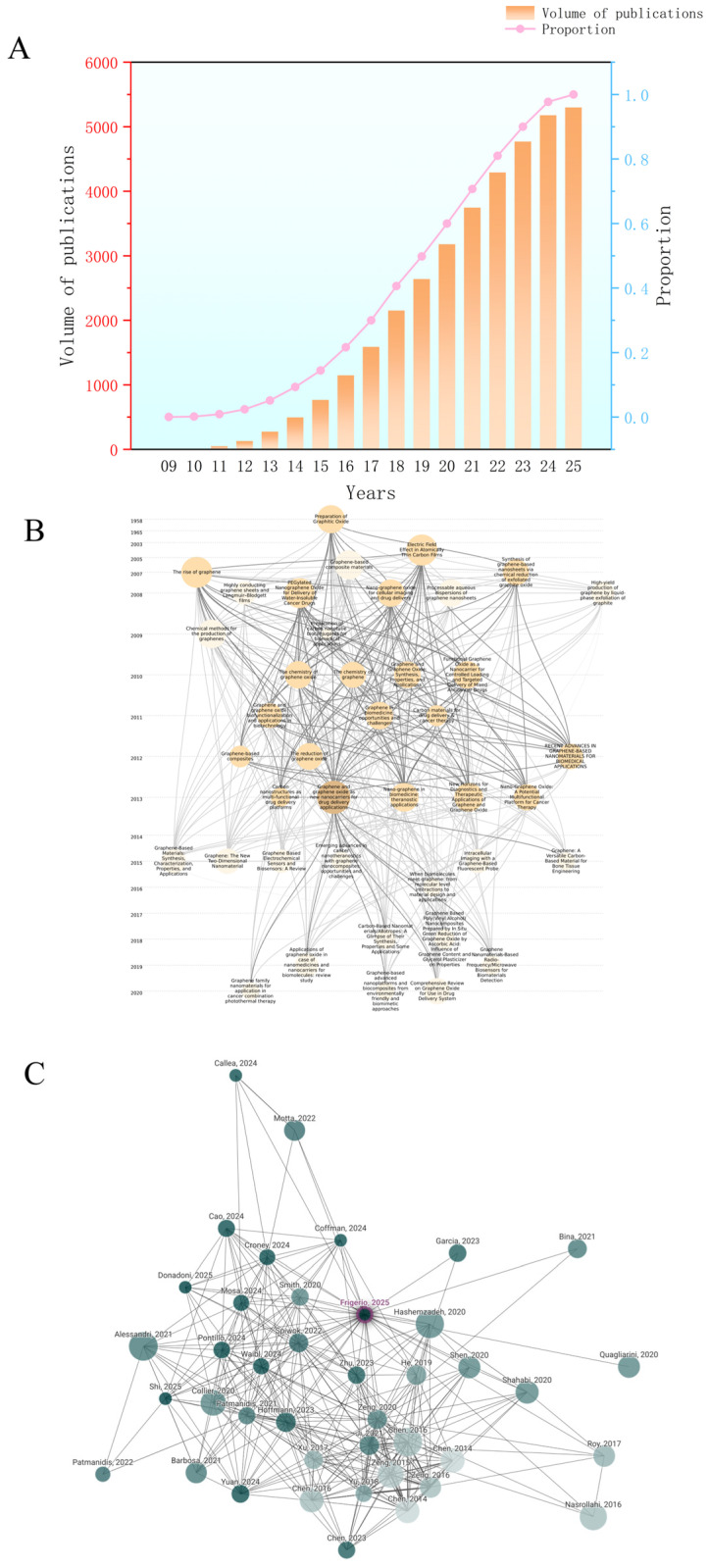
Research trends (**A**) and scientific knowledge network analysis (**B**,**C**) of graphene applications in the biomedical field. Figure 1A illustrates the number of publications related to the field from 2009 to 2025. In 2009, very few articles were published in this domain. However, since 2014, there has been a rapid increase in the number of publications, indicating the growing interest and development in this research area. Figure 1B,C depict the citation relationships among key publications related to the application of GFNs in drug delivery systems from 2004 to 2025. Each node represents a distinct research paper, with the size of the node reflecting the frequency with which the paper has been cited by others. The lines connecting the nodes indicate citation relationships [3,4,9].

**Figure 2 pharmaceuticals-18-01245-f002:**
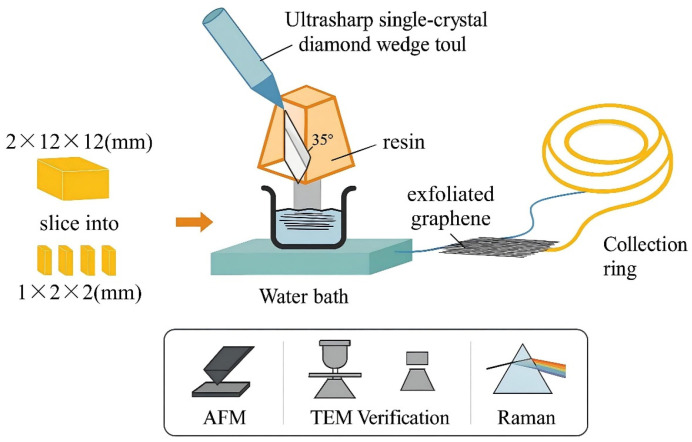
Schematic diagram of the mechanical exfoliation process for preparing GO. Graphite blocks (2 × 12 × 12 mm) are cut, embedded in resin, and exfoliated in a water bath using a 35° diamond wedge tool. Exfoliated graphene is collected and characterized via AFM, TEM, and Raman spectroscopy.

**Figure 3 pharmaceuticals-18-01245-f003:**
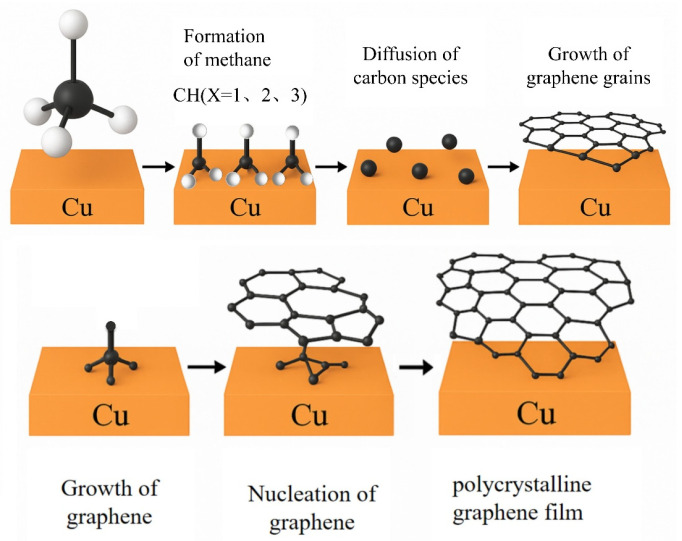
Schematic of graphene growth on Cu by CVD: Methane decomposes on the Cu surface to form carbon species, which diffuse, nucleate, and grow into graphene grains, eventually forming a polycrystalline graphene film. Black circles represent carbon atoms, and white circles represent hydrogen atoms.

**Figure 4 pharmaceuticals-18-01245-f004:**
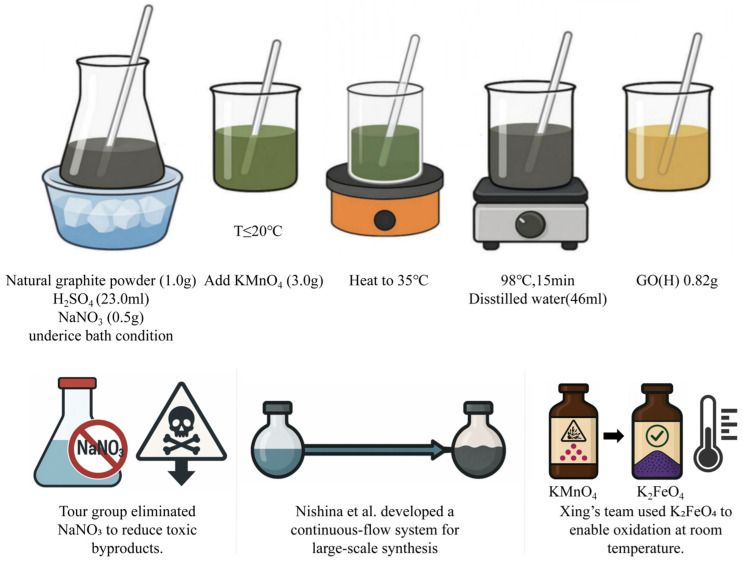
Schematic diagram of graphene oxide synthesis via the Hummers method and its improved variants. Graphite, H_2_SO_4_, and NaNO_3_ are mixed under ice bath; KMnO_4_ is added below 35 °C; after dilution with water, the mixture is heated to 98 °C for 15 min, yielding graphene oxide dispersion.

**Figure 5 pharmaceuticals-18-01245-f005:**
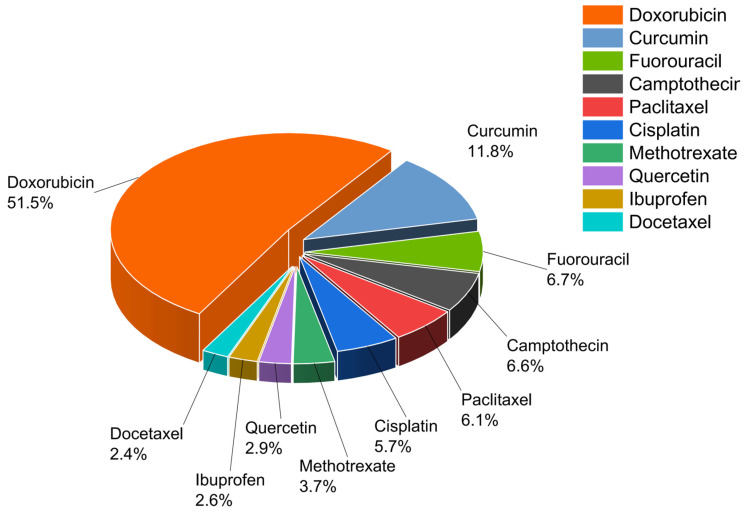
This pie chart illustrates the types and proportions of drugs commonly used in GFN-based drug delivery research. Doxorubicin is the most frequently loaded drug, accounting for 51.5%. This is followed by Curcumin (11.8%), Fluorouracil (6.7%), Camptothecin (6.6%), and Paclitaxel (6.1%), all of which are representative anticancer drugs.

**Table 1 pharmaceuticals-18-01245-t001:** GFNs-Based drug delivery systems for breast cancer: delivered drugs, addressed delivery issues, and study highlights.

Carrier	Method; Size; and Zeta Potential	Drug	Addressed Drug Delivery Issue	Study Highlight	Reference
FA-GO	Hummer’s Method; 1070 nm; −29 mV	MTX, DOX	The conventional release of MTX is too rapid, with poor targeting and significant side effects.	Construction of a multifunctional nanoplatform with a clearly defined targeting mechanism and good biocompatibilit.	[72]
PAC@HA-ADH-GQDs System	GQDs were synthesized from citric acid. HA–ADH was prepared from HA, ADH, and EDC, HA–ADH–GQDs were formed in PBS; 23 ± 2.62 nm; −26.5 ± 1.32 mV	PTX	Free paclitaxel is poorly water-soluble, requires toxic solvents, causes side effects and releases prematurely in normal tissues.	Specific binding of HA to CD44 receptors enhances targeting; improves PTX anticancer activity.	[47]
GO	Modified Hummers’ method; Unavailable; About −30 mV	Oligonu-cleotide Drugs	Difficult to control drug release, lack of responsive mechanisms, oligonucl-eotide drugs are prone to degradation and have low delivery efficiency.	The system can deliver drugs while monitoring the surrounding pH environment in real time, demonstrating potential for “heranostic integration”.	[118]
GO-TD-Fe_3_O_4_@PEG System	Hummer’s Method; 144.21 nm; −51.1 mV	DOX	Conventional DOX has high toxicity, poor targeting, and nanoparticle aggregation issues.	Magnetic targeting capability, excellent biocompatibility and stability.	[119]
GO-PAMAM System	GO was made using a modified Hummer’s method, then linked to PAMAM dendrimer via EDC/NHS coupling; Unavailable; Unavailable	Quercetin	Poor solubility and low delivery efficiency of hydrophobic drugs; poor targeting and low drug utilization.	Developed a pH-sensitive smart nanocarrier; improved solubility and stability of hydrophobic drugs.	[50]
DOX@GNFH	Modified Hummer’s method for GO; FA and HZ-PEG conjugation via EDC/NHS amidation; 110 and 172 nm; Unavailable	DOX	Lack of tumor targeting, normal tissue easily affected; limited drug solubility and stability.	Constructed a multifunctional integrated nano delivery platform; FA-mediated active targeting.	[120]
MGO@APT Carrier	GO functionalized with β-CD and FA via CDI activation and covalent coupling; Unavailable; −20.4 mV	PTX	Poor water solubility of paclitaxel, rapid metabolic clearance; uncontrollable release, significant side effects.	Dual enhancement of targeting and imaging capabilities, efficient PTX loading and controlled release.	[99]
HA-GO/Fe_3_O_4_ Nanocomposite	GO was prepared by modified Hummers method, HA grafted via EDC/NHS coupling, and Fe_3_O_4_ nanoparticles added to form the composite; 166.8 ± 16.2 nm; −21.5 ± 2.25 mV	PTX, DOX	Limited effectiveness of single treatment modality; inability to deliver multiple drugs simultaneously.	Constructed a multifunctional synergistic treatment platform; combined drug co-delivery for synergistic therapy.	[121]

Abbreviations: Folic Acid–Graphene Oxide, FA-GO; Methotrexate, MTX; Doxorubicin, DOX; Paclitaxel loaded on Hyaluronic Acid–Adipic Dihydrazide–Graphene Quantum Dots, PAC@HA-ADH-GQDs;Paclitaxel, PTX; Hyaluronic Acid, HA; Cluster of Differentiation 44, CD44; Graphene Oxide, GO; Graphene Oxide–Thiodiglycol–Iron Oxide coated with Polyethylene Glycol, GO-TD-Fe_3_O_4_@PEG; Polyethylene Glycol, PEG; Magnetite (Iron(II,III) oxide), Fe_3_O_4_; Graphene Oxide–Polyamidoamine, GO-PAMAM; CDI-activated β-CD and FA co-modified graphene oxide nanocarriers, GO–CD–FA; Aptamer, APT; Hyaluronic Acid-modified Graphene Oxide/Magnetite Nanocomposite, HA-GO/Fe_3_O_4_.

**Table 2 pharmaceuticals-18-01245-t002:** GFNs-Based drug delivery systems for lung cancer: delivered drugs, addressed delivery issues, and study Highlights.

Carrier	Method; Size; and Zeta Potential	Drug	Addressed Drug Delivery Issue	Study Highlight	Reference
Hollow Graphene Oxide Microcapsules	GO was made by the improved Hummers method, then freeze-dried after liquid nitrogen treatment to obtain hollow GOMs; 0.5–3.5 μm; −42.5 mV	Hydrophobic Anticancer Drugs	Low cellular uptake rate, poor membrane permeability, poor solubility of hydrophobic drugs.	Applicable to various hydrophobic drugs; high cellular uptake and rapid intracellular delivery capacity.	[43]
GO–PLA–PEG Nano System	PLA–PEG was synthesized by ring-opening polymerization and used to form NGO/PLA–PEG composites via sonication and centrifugation; 534.1 ± 9.24 nm; Unavailable	PTX	Poor water solubility of paclitaxel, requires toxic solubilizing agents.	Efficient PTX loading; enables controlled release.	[123]
NGO-SS-HA Nano Delivery System	Conjugation of hyaluronic acid to nano-graphene oxide via disulfide linkers; 125 nm; Unavailable	Gefitinib	Difficult delivery and low bioavailability of water-insoluble drugs; non-targeted release leads to significant side effects.	Constructed a GSH-responsive targeted delivery system; enables delivery of water-insoluble drugs.	[124]

Abbreviations: poly(lactide), PLA; Polyethylene Glycol, PEG; Paclitaxel, PTX; Nano Graphene Oxide–Disulfide bond–Hyaluronic Acid, NGO-SS-HA; Glutathione, GSH.

**Table 3 pharmaceuticals-18-01245-t003:** GFNs-Based drug delivery systems for cervical cancer: delivered drugs, addressed Delivery issues, and study highlights.

Carrier	Method; Size; and Zeta Potential	Drug	Addressed Drug Delivery Issue	Study Highlight	Reference
GO@LM-SP-FA System	GO@LM-SP-FA was prepared by EDC/NHS-mediated conjugation of LM-SP-FA to GO, followed by DOX loading via non-covalent interactions; 257 nm; +5.3 mV	DOX	Hydrophobic anticancer drugs are difficult to dissolve, with low delivery efficiency.	Constructed a dual-responsive (pH and photothermal) smart drug delivery system; possesses photothermal therapy potential.	[125]
FGO	Ultrasmall FGO was prepared by alkali activation, mild oxidation, and sonication, then functionalized with folic acid.; ~50 nm; Unavailable	Anticancer Drugs	Drug carriers lacking tracking capability; single-function carriers without responsive control.	Ultra-small material size; excellent fluorescence performance and switchable control capability.	[126]
GO-PEG-EGFR	Ultrasmall FGO (~50 nm) was prepared by alkali activation, oxidation, and folic acid conjugation.; 1000 nm; −30.72 mV	GAD	Natural anticancer compound GAD has poor water solubility and low bioavailability; traditional chemotherapy drugs have high toxicity and severe drug resistance issues.	Innovatively utilized natural anticancer compound GAD combined with functionalized GO as a delivery platform.	[53]
GO–HA	Prepared by EDC/NHS-mediated ADH amination of GO followed by hyaluronic acid conjugation; 40–350 nm; −34.2 mV	Anticancer Drugs	Poor carrier stability, prone to aggregation or rapid degradation in vivo; insufficient functional sites, making it difficult to load multiple drugs or functional factors simultaneously.	Introduction of amino functional groups to enhance targeting ability and increase the number of active sites, equipped with CD44 targeting capability.	[127]

Abbreviations: Graphene Oxide@Lipid Membrane-Silibinin-Polydopamine-Folic Acid, GO@LM-SP-FA System; Doxorubicin, DOX; Fluorinated graphene oxide, FGO; Graphene Oxide, GO; Ganoderenic acid D, GAD; Cluster of Differentiation 44, CD44; Graphene oxide-polyethylene glycol-anti-epidermal growth factor receptor, GO-PEG-EGFR; Hyaluronic acid, HA.

**Table 4 pharmaceuticals-18-01245-t004:** GFNs-Based drug delivery systems for liver cancer: delivered drugs, addressed delivery issues, and study highlights.

Carrier	Method; Size; and Zeta Potential	Drug	Addressed Drug Delivery Issue	Study Highlight	Reference
GYSMNP@PF127 System	GO–Fe_3_O_4_ synthesized via co-precipitation; PVP/PAA grafted via polymerization; 180.0 ± 25.4 nm; −36.8 ± 2.0 mV	DOX	Uncontrolled drug release, narrow therapeutic window; limited anticancer effect with single therapy.	Dual stimulus-responsive capability; magnetic targeting.	[128]
GON/CS/CS-DMMA System	Prepared by electrostatic self-assembly of GO with chitosan and DMMA-chitosan; 181 nm; −26.5 mV	DOX	Premature drug release in normal tissues causing side effects.	The drug delivery system features pH responsiveness and charge-switching capability.	[129]
GO–PEI–PEG/CS-Aco/shRNA	Fabricated via layer-by-layer self-assembly of functional polymers and therapeutic agents on GO nanosheets; 100–250 nm; 10.97 ± 0.72 mV	shABCG, DOX	Severe chemotherapy resistance in liver cancer; instability and delivery difficulty of siRNA/gene drugs.	Co-delivery of chemotherapy drugs and siRNA; layered structure enables simultaneous loading of both molecules.	[130]
LA–CMC–GO Nanocomposite System	GO prepared via modified Hummers’ method, functionalized with CMC, FI, and LA, acetylated; Unavailable; −40.1 ± 1.2 mV	DOX	Poor carrier water solubility, insufficient biocompatibility; premature drug release in non-target areas.	Developed a dual-functionalized graphene oxide nanocarrier; enhances anticancer efficacy while reducing systemic toxicity.	[131]
HA-Modified Graphene Oxide Nanocomposite (GO–HA)	DOX was noncovalently loaded onto GO and coated with ADH-modified hyaluronic acid to form HA–GO–DOX; ~250 nm; Unavailable	DOX	Premature drug release in normal tissues reduces therapeutic efficiency; lack of targeting leads to significant side effects.	Active targeting achieved Via HA recognition of CD44 receptors on tumor cell surfaces.	[132]
Heparin-Modified GO	GO was carboxylated, ADH-conjugated, heparin-modified, and loaded with DOX; 119.8 nm; −39 mV	DOX	Poor water solubility, Uncontrolled drug release.	Multi-functional integrated platform, Highly versatile and adaptable	[133]

Abbreviations: hydrophilic graphene-based yolk-shell magnetic nanoparticles functionalized with copolymer pluronic F-127, GYSMNP@PF127; Chitosan–Aconitic Anhydride, CS-Aco; Polyethylene Glycol, PEG; Polyethylenimine, PEI; Short Hairpin RNA, shRNA; carboxymethyl chitosan, CMC; lactobionic acid, LA; Chitosashn, CS; Carboxymethyl Chitosan, CMCS; Lactic Acid–Carboxymethyl Chitosan–Graphene Oxide, Lac-CMC-GO; Dimethylmaleic Anhydride, DMMA; Short hairpin RNA against ATP-binding cassette sub-family G member 2, shABCG; Graphene Oxide–Hyaluronic Acid, GO–HA.

**Table 5 pharmaceuticals-18-01245-t005:** GFNs-Based drug delivery systems for other cancers: delivered drugs, addressed delivery issues, and study highlights.

Cancer Type	Carrier	Method; Size; and Zeta Potential	Drug	Addressed Drug Delivery Issue	Study Highlight	Reference
OSCC	NPF@DOX.	DOX-loaded nano-graphene oxide was functionalized with PEG and conjugated with a FAP-targeting peptide; Unavailable; Unavailable	DOX	Limited efficacy of single treatment modality, prominent drug resistance issues.	Features dual therapeutic mechanisms: pH responsiveness and photothermal conversion.	[97]
OSCC	NGO-BBN-AF750	NGO conjugated with BBN-AF750; via mixing and centrifugation; 0.5–5 μm; −16.6 mV	DOX	Premature drug release at normal pH, low efficiency.	Dual-functional platform design; significantly enhances anticancer activity of the drug in OSCC.	[103]
Osteosarcoma	pH-Responsive Functionalized GO Nano System	Prepared by TRIS modification of GO in DMF, hydroxyl activation under alkaline conditions, QIFO coupling, followed by DOX loading; Unavailable; Unavailable	DOX	Premature DOX release in normal tissues, significant side effects.	Simple structural design with clear functionality; complete material characterization providing a reliable reference for future studies.	[134]
Malignant Tumors with Acidic and Highly Reductive Microenvironment Characteristics	MGOMCs	Prepared by coupling cysteine-functionalized GO with oleic acid-modified Fe_3_O_4_ via high-intensity sonication; 2.0 μm; Unavailable	Hydrop-hobic Chemot-herapy Drugs, such as DOX, PTX		First to form microcapsule structure by polymer coating Fe_3_O_4_ with GO; possesses smart release capability.	[135]

Abbreviations: Nano-drug delivery system, NPF@DOX; Oral Squamous Cell Carcinoma, OSCC; Graphene Oxide, GO; Nano-graphene oxide, NGO; bombesin antagonist peptides conjugated to Alexa Fluor 750, BBN-AF750; Magnetic graphene oxide microcapsules, MGOMCs.

**Table 6 pharmaceuticals-18-01245-t006:** GFNs-Based drug delivery systems for AD and PD: delivered drugs, addressed delivery issues, and study highlights.

Disease Type	Carrier	Method; Size; and Zeta Potential	Drug	Addressed Drug Delivery Issue	Study Highlight	Reference
AD	GO	GO dispersion was stirred with Dau solution for 24 h, then ultrafiltered to remove free drug; 250.91 ± 15.16 nm; −19.8 ± 0.72 mV	Dauricine	Traditional oral dauricine has low absorption and is easily metabolized and inactivated.	First to load dauricine onto GO for intranasal delivery.	[136]
AD	CS–GQDs	Prepared by microfluidic electrostatic crosslinking of chitosan and graphene quantum dots with sodium tripolyphosphate; 10–20 nm; −96.3 mV		Conventional carriers have large particle sizes and non-specific biodistribution.	achieved non-invasive intranasal delivery to cross the BBB.	[137]
AD	Clitoria ternatea GQDs	Prepared via one-step microwave-assisted synthesis from Clitoria ternatea flower extract; 10 ± 1.3 nm; −46 ± 0.4 mV		Traditional GQDs synthesis methods have high toxicity and are not environmentally friendly; neurotoxicity induced by β-amyloid is difficult to reverse.	Green synthesis of GQDs; exhibits strong antioxidant properties and biological safety.	[138]
AD	Antibody Functionalized GO Fluorescent Probe Sensing System	Prepared by EDC/NHS-mediated covalent coupling of antibodies to carboxyl groups on graphene oxide; Unavailable; Unavailable		Lack of rapid and sensitive tau protein detection methods.	high selectivity and specificity.	[139]
AD	N-GQDs	N-GQDs were synthesized hydrothermally from citric acid and ammonia, followed by pH adjustment and dialysis; 4.6 ± 0.78 nm; Unavailable	Tacrine	Traditional drug detection methods are complex, time-consuming, and costly.	Regulate N-GQDs fluorescence intensity using enzymatic reaction products.	[140]
AD	Nitrogen-doped Graphene Quantum Dots, N-GQDs	Microwave synthesis from glucose and ammonia, pH adjustment, dialysis; 5–10 nm; −1.06 mV	PC-25, PC-37, PC-48, Tacrine	Challenges in High-Throughput Screening and Enzyme Inhibition Efficiency Monitoring for Anti-AD Drugs.	Established a simple monitoring system for drug-enzyme interaction.	[141]
PD	GO	Prepared by loading puerarin onto GO, PEGylation, and covalent conjugation with lactoferrin via EDC/NHS coupling; 236.1 nm; −24.74 mV		PD drugs have difficulty crossing the blood–brain barrier.	traditional carriers lack neuroprotective function First application of the GO platform for PD treatment.	[142]
PD	GO Sheet Materials with Different Microstruct-ures	One-step confined hydrothermal synthesis of rGO microfibers; Unavailable; Unavailable		High toxicity and uncertainty of nanomaterials in the nervous system; drug delivery platforms struggle to form effective interfaces with neural cells.	Demonstrated good biocompatibility of GO with the nervous system.	[143]

Abbreviations: Alzheimer’s Disease, AD; Graphene Oxide, GO; Chitosan–Graphene Quantum Dots, CS–GQDs; Clitoria ternatea Mediated Graphene Quantum Dots, Clitoria ternatea–GQDs; Graphene Quantum Dots, GQDs; Blood–Brain Barrier, BBB; Nitrogen-Doped Graphene Quantum Dots, N–GQDs; Parkinson’s disease, PD.

**Table 7 pharmaceuticals-18-01245-t007:** GFNs-based cardiovascular diseases delivery systems: delivery, delivery issues addressed, and research priorities.

Disease Type	Carrier	Method; Size; and Zeta Potential	Drug	Addressed Drug Delivery Issue	Study Highlight	Reference
Myocardial Infarction	Vascularized Conductive Elastic Patch	A vascularized conductive elastic scaffold was prepared by cryopolymerizing holey graphene oxide/polypyrrole in a poly(hydroxyethyl methacrylate) matrix; Unavailable; Unavailable		Existing myocardial repair methods cannot effectively guide angiogenesis.	The material is composed of conductive polymers, elastic biopolymers, and extracellular matrix composites, combining biocompatibility with functionality.	[144]
Myocardial Infarction	Injectable Silk Fibroin-GO Hydrogel	The clustery GO–silk fibroin composite was obtained by self-assembly, freeze-drying, and crosslinking; Unavailable; Unavailable		Low cell viability in cell delivery	GO was first combined with silk fibroin to create a conductive injectable hydrogel, offering a safe and efficient delivery system for myocardial infarction treatment.	[145]
Atheroscle- rosis	Gelatin-Functionalized GO Nano Platform	Obtained by gelatin–GO conjugation and π–π loading of atorvastatin; 250–400 nm; −17.67 ± 1.9 mV	Atorvastatin	Traditional drugs face low bioavailability and high-dose side effects.	Innovative drug delivery system design; effectively promotes lipid efflux.	[146]
Hypertension and Heart Disease	HyA/Gel/PEO	Electroconductive HyA/Gel/PEO films with RGO were prepared by solvent casting with EDC crosslinking; Unavailable; Unavailable	Irbesartan	Uncontrollable or inaccurate drug release process.	Established a mathematical model to predict drug release behavior in rGO-hydrogel systems.	[147]
Coronary Heart Disease	PEI-PEG@GO	Prepared by EDC-mediated amide coupling of PEI and PEG to GO, then dip-coated onto PLA stents; Unavailable; Unavailable	S-nitrosoglutathione S-nitroso-N-acetylpenicillamine	Short-acting and uncontrollable NO release; drug-eluting stents delay endothelial repair, leading to thrombosis.	First development of amine-functionalized graphene coating.	[148]
Complicatio-ns Caused by Atheroscler-osis	SNO-Cys@PGO	EDC/NHS-mediated cysteamine grafting onto porous GO, followed by NO loading with acidified nitrite; Unavailable; Unavailable	NO	Poor NO release sustainability; partial carrier degradation remains incomplete.	Dual-mechanism controlled NO release.	[149]

Abbreviations: Polyethyleneimine–Polyethylene Glycol Grafted Graphene Oxide, PEI–PEG@GO; Nitric Oxide, NO; hyaluronic acid, HyA; gelatin, GEL; poly(ethylene oxide), PEO; S-nitrosoglutathione, GSNO; S-nitrosocysteamine-functionalised, SNO-Cys@PGO porous graphene oxide, PGO.

**Table 8 pharmaceuticals-18-01245-t008:** GFNs-based bacterial infections delivery systems: delivery, delivery issues addressed, and research priorities.

Disease Type	Carrier	Method; Size; and Zeta Potential	Drug	Addressed Drug Delivery Issue	Study Highlight	Reference
infected wounds	GO/AgNW-assisted Starch/PVA nanocomposite film	GO-AgNWs was obtained by ultrasonication of GO and AgNWs for 20 min; 98.9 nm and 773.8 nm; −0.6 mV	CIP	Traditional dressings lack controlled release capability, exhibit poor drug uniformity, and allow rapid diffusion.	A multifunctional dressing platform was constructed with enhanced mechanical strength.	[150]
Chronic or Acute Wound Infection	Chitosan/PVA/GO Electrospun Nanofiber Membrane	CS/PVA/GO/CIP nanofibrous membrane was prepared by electrospinning a mixed solution of CS, PVA, GO, and ciprofloxacin; Unavailable; Unavailable	CIP	Traditional wound dressings have short drug release duration and limited antibacterial time.	The incorporation of GO enhances mechanical strength and antibacterial synergistic effects.	[67]
Bacterial Infection	Methanol-Derived Graphene	Gentamicin was loaded onto methanol-derived graphene by sonication and stirring at pH 7; Unavailable; Unavailable	Gentamicin Sulfate	Improved controlled-release efficiency of the antibiotic gentamicin sulfate, reduced burst release.	high drug loading capacity up to 2.57 mg/mg; clear drug release mechanism.	[64]
Bacterial Infection	Fe_3_O_4_@GO Magnetic GO Nanocomposite	MGO was synthesized by loading Fe_3_O_4_ NPs onto GO; then GS was grafted via amidation to obtain GS-MGO nanohybrids; Unavailable; Unavailable	Gentamicin Sulfate	Antibiotics easily inactivate, insufficient sustained release.	First exploration of a high-stability strategy combining antibiotics with GO Via covalent grafting.	[65]
Pseudom-onas aerugino-sa Infection	rGO	Prepared by mixing rGO with ciprofloxacin hydrochloride; 120 nm; +7.5 mV	CIP	CIP exhibits strong resistance and low efficiency in bacterial infection treatment.	Constructed an efficient antibacterial nanosystem; rGO physically damages bacterial membranes, synergizing with CIP’s antibacterial mechanism to enhance overall efficacy.	[151]
Bone Tissue Injury or ImplantAssociat-ed Infection	GO/Polydopamine Functionalized Porous PEEK Surface Coating System DEX	SP plates were coated with polydopamine and modified with GO and Dex-loaded liposomes; Unavailable; Unavailable	DEX	Traditional orthopedic implant materials have poor bioactivity and are prone to infection.	The modified layer provides dual biological functions of antibacterial and osteogenic activity.	[71]

Abbreviations: Silver Nanowire, AgNW; Polyvinyl Alcohol, PVA; Ciprofloxacin, CIP; Iron Oxide, Fe_3_O_4_; Chitosan, CS; magnetic graphene oxide, MGO; Dexamethasone, DEX; Polyetheretherketone, PEEK.

**Table 9 pharmaceuticals-18-01245-t009:** GFNs-based inflammation delivery systems: delivery, delivery issues addressed, and research priorities.

Disease Type	Carrier	Method; Size; and Zeta Potential	Drug	Addressed Drug Delivery Issue	Study Highlight	Reference
Pneumonia	nGO	Prepared by ultrasonically dispersing GO in water, mixing with dexamethasone for adsorption, and drying; 150 nm; +20 mV to +30 mV	DEX	DEX as a hydrophobic drug, has low solubility, poor absorption, and low bioavailability in the body.	Improved the bioavailability of orally administered drugs; maintained the crystalline properties of the drug.	[70]
Inflammation	GO–CS Material	Prepared by dispersing GO in chitosan solution, mixing to form a homogeneous composite, and drying; Unavailable; Unavailable	DEX-P	Enhanced the carrier’s controlled-release performance, reduced toxicity, and prolonged therapeutic efficacy.	voiding high toxicity and extending drug efficacy Significantly suppressed burst drug release, demonstrating good cell compatibility and thermal stability.	[152]
Chronic Viral Infection	GO	Prepared by mixing ~20 nm graphene oxide with rat anti-mouse IL-10R antibodies, then washing with PBS; Unavailable; Unavailable	Anti-IL-10 Receptor Antibo-dy	Poor antibody stability, short release duration, insufficient targeting.	First demonstration of GO as a drug delivery platform for anti-IL10R Ab; strong sustained-release capability.	[153]

Abbreviations: Nano Graphene Oxide, nGO; Chitosan, CS; Dexamethasone, DEX; Dexamethasone Phosphate, DEX-P; Interleukin-10 Receptor, IL-10R;Antibody, Ab.

**Table 10 pharmaceuticals-18-01245-t010:** GFN-based diabetes delivery systems: delivery, delivery issues addressed, and research priorities.

Carrier	Method; Size; and Zeta Potential	Drug	Addressed Drug Delivery Issue	Study Highlight	Reference
PEGDMA-rGO	Prepared by incorporating reduced graphene oxide into PEGDMA hydrogel; an area of 1 cm^2^; Unavailable	Insulin	Conventional insulin delivery systems face issues with sustained release and lack of stimulus responsive capability.	Incorporation of intelligent response mechanisms to achieve on-demand insulin release.	[154]
GO-AuNPs	prepared by in situ chemical reduction of HAuCl_4_ on ultrasonically dispersed GO; Unavailable; Unavailable	Insulin	Optimizing Drug Nanomaterial Interactions.	Synthesis of various GO–Au composite materials and systematic comparison.	[80]
In@GO NgC	prepared by mixing GO nanogels with insulin for molecular intercalation; 11.0 nm; Unavailable	Insulin	Oral insulin delivery is challenging.	Achieved a breakthrough in oral insulin delivery; developed a pH-responsive controlled-release system.	[83]
RGO/Ni(OH)_2_ film	rGO/Ni(OH)_2_ films prepared by cathodic EPD after Ni^2+^-induced zeta potential reversal, then loaded with insulin; Unavailable; +14 mV	Insulin	Conventional insulin therapy lacks real-time feedback mechanisms.	Integrated “sensing–release” dual-function system; electrochemically triggered controlled insulin release.	[84]
GO	Synthesized through ultrasonication; 4 μm; –27.5 mV	Insulin	Insulin is easily degraded by gastric acid, with extremely low oral bioavailability.	Improved insulin stability and bioavailability.	[155]

Abbreviations: rGO impregnated poly(ethylene glycol) dimethacrylate based hydrogels, PEGDMA-rGO; Gold Nanoparticles, AuNPs; An insulin intercalated GO based nanogel composite, In@GO NgC.

**Table 11 pharmaceuticals-18-01245-t011:** Toxicological Profiles of GFNs via Different Administration Routes.

Carrier Type	Method of Administration	Main Accumulation Site	Toxicity Manifestations	Reference
GO	Intratracheal instillation	Lungs (primary), small amounts in liver, intestine	Dose- and time-dependent acute lung injury (ALI) characterized by increased LDH, increased ALP, protein leakage, pulmonary edema, and neutrophil infiltration; Persistent retention in lungs for up to 3 months, leading to chronic pulmonary fibrosis and collagen deposition	[166]
GO	intratracheal instillation	Lungs (alveolar regions, airways for aggregated GO)	Oxidized GO induced severe and persistent lung injury, mitochondrial ROS generation, inflammation, and apoptosis; nanoscale-dispersed pristine graphene exhibited markedly reduced toxicity	[167]
PEG-GO	Oral	Gastrointestinal	No significant acute or chronic toxicity; normal hematology, biochemistry, and histopathology	[166]
PEG-GO	Intraperitoneal	Liver, spleen, kidney, lung	No significant organ damage despite high tissue accumulation; normal hematology and biochemistry	[166]
GO	Intravenous injection	liver, spleen, kidneys, and embryonic tissues.	Increased embryo resorption, decreased survival rate, developmental delay; alterations in maternal gut microbiota and metabolites associated with inflammation and metabolic disruption.	[168]
GO	nose inhalation	Alveolar regions of the lungs	High concentration (1.88 mg/m^3^) caused mild alveolitis and inflammation on day 1, which largely resolved by day 14 without systemic toxicity	[169]

Abbreviations: Polyethylene glycol-functionalized graphene oxide, PEG-GO; Lactate Dehydrogenase, LDH; Alkaline Phosphatase; ALP.

**Table 12 pharmaceuticals-18-01245-t012:** Summary of Toxicological Profiles of Functionalized Graphene Oxide Variants.

Functional Group/Material	Toxicity Type	Dose	Reference
Pristine GO(Epoxy group, Hydroxyl group)	Platelet aggregation via Src kinase activation, calcium release, ROS generation, cytoskeletal changes, mitochondrial potential collapse, apoptosis (high concentration)	0.5–20 μg/mL in vitro; 250 μg/kg in vivo (mice)	[171]
Pristine GO(Epoxy group, Hydroxyl group)	pristine graphene oxide	10–100 μg/mL	[170]
rGO/reduced oxygen functional groups, altered charge distribution	Minor platelet aggregation, weaker activation, minimal thrombus induction	2–10 μg/mL in vitro; 250 μg/kg in vivo (mice)	[171]
Carboxyl-functionalized GO (GO-COOH)	Lower hemolysis rate, reduced oxidative damage compared to GO	25 μg/mL	[170]
Polyethylenimine-functionalized GO, (GO-PEI)	Severe hematotoxicity to T lymphocytes via membrane damage		[171]
Amine-modified graphene (G-NH_2_)	No thrombogenic or platelet-activating effects; does not compromise red blood cell integrity.	250 μg/kg body weight	[172]

## Data Availability

Not applicable.

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
