# Peer review of "Advances in the Application of Graphene and Its Derivatives in Drug Delivery Systems"

_pharmaceuticals, 2025, doi:10.3390/ph18091245_

Round 1
Reviewer 1 Report
Comments and Suggestions for Authors
Dear Editor,
I have reviewed the article titled "Advances in the Application of Graphene and Its Derivatives in Drug Delivery Systems’. This review systematically outlines the latest research advancements regarding graphene and its derivatives in drug loading, targeted delivery, and smart release. Below are some suggestions for authors that can improve the manuscript.
The Tour group eliminated NaNO₃ to reduce the risk of toxic byproducts [27], Nishina et al. developed a continuous-flow reaction system to accommodate large-scale production [28], and Xing's team replaced KMnO₄ with K₂FeO₄, enabling the oxidation reaction to proceed at room temperature [29]. Add these methods in graphical form.
2.2 Drug Loading of GFNs: Lines 282-318, Add the relevant references of supported literature citation.
2.2.1 Delivery of Hydrophobic Small-Molecule Drugs: Lines 320-365, Add the relevant references of supported literature citation.
2.2.2 Delivery of Hydrophilic Small-Molecule Drugs: Lines 395-425, Add the relevant references of supported literature citation.
2.2.3 Delivery of Macromolecular Drugs: Lines 455-476, Add the relevant references of supported literature citation.
2.3 Targeted Delivery of GFNs: Add some examples with references.
3.1.1 Breast Cancer: Lines 601-620, Add the relevant references of supported literature citation.
Add the relevant abbreviations full form under the table of respective tables.
Add the extra column to add the method of preparation, size, and zeta potential of drug-loaded GO formulations in all tables.
Add clinical research and the patented formulation of Graphene and Its Derivatives in Drug Delivery Systems.
Add recent years (2025) research literature.
Comments on the Quality of English Language
In many places, run-on sentences need to be rewritten.
Check the manuscript for grammatical, typo errors.
Author Response
Comments 1: The Tour group eliminated NaNO₃ to reduce the risk of toxic byproducts [27], Nishina et al. developed a continuous-flow reaction system to accommodate large-scale production [28], and Xing's team replaced KMnO₄ with K₂FeO₄, enabling the oxidation reaction to proceed at room temperature [29]. Add these methods in graphical form.
Response 1: Thank you for pointing this out. We agree with this comment. Therefore we have added three important improvement strategies to Figure 1. These additions make the illustration more comprehensive and intuitive in reflecting the major advancements in this field. The revised graphical content appears on Page 9, Line 273 (Figure 4). We hope this modification meets the reviewer’s expectations.
Comments 2:2.2 Drug Loading of GFNs: Lines 282-318, Add the relevant references of supported literature citation.
Response 2: Thank you for pointing this out. We agree with this comment. Therefore we we have added the relevant supporting references to Section 2.2: Drug Loading of GFNs. Specifically, citations [30]–[36] have been included to strengthen the discussion and provide literature support. The updated content is located on Page 10, Lines 278–315.
Comments 3: 2.2.1 Delivery of Hydrophobic Small-Molecule Drugs: Lines 320-365, Add the relevant references of supported literature citation.
Response 3: Thank you for pointing this out. We agree with this comment. Therefore we have incorporated the relevant supporting references into Section 2.2.1: Delivery of Hydrophobic Small-Molecule Drugs. Specifically, citations [31], [37]–[47] have been added to enhance the scientific support of this section. The revised content is located on Page 11, Lines 317–354.
Comments 4: 2.2.2 Delivery of Hydrophilic Small-Molecule Drugs: Lines 395-425, Add the relevant references of supported literature citation.
Response 4: Thank you for pointing this out. We agree with this comment. Therefore We have added the relevant supporting references to Section 2.2.2: Delivery of Hydrophilic Small-Molecule Drugs to strengthen the discussion. Specifically, references [56]–[64] have been included. The revised content is located on Page 12, Lines 392–422.
Comments 5: 2.2.3 Delivery of Macromolecular Drugs: Lines 455-476, Add the relevant references of supported literature citation.
Response 5: Thank you for pointing this out. We agree with this comment. Therefore we have added relevant supporting references to Section 2.2.3: Delivery of Macromolecular Drugs to improve the scientific foundation of the discussion. Specifically, references [74]–[80] have been cited. The updated content is located on Page 13, Lines 452–473.
Comments 6: 2.3 Targeted Delivery of GFNs: Add some examples with references.
Response 6: Thank you for pointing this out. We agree with this comment. Therefore we have added specific examples along with corresponding references to Section 2.3: Targeted Delivery of GFNs to enhance the clarity and depth of the discussion. The added references are [92]–[96], and the revised content appears on Page 15, Lines 528–544.
Comments 7: 3.1.1 Breast Cancer: Lines 601-620, Add the relevant references of supported literature citation.
Response 7: Thank you for pointing this out. We agree with this comment. Therefore We have added the relevant supporting references to Section 3.1.1: Breast Cancer to strengthen the discussion. Specifically, references [112]–[118] have been incorporated. The revised content can be found on Pages 17–18, Lines 611–642.
Comments 8: Add the extra column to add the method of preparation, size, and zeta potential of drug-loaded GO formulations in all tables.
Response 8: Thank you for pointing this out. We agree with this comment. Therefore we have added an additional column titled “Method; Size; Zeta Potential” to all relevant tables to provide a more comprehensive comparison of drug-loaded GO formulations. For entries where the required data were not available in the original publications, we have indicated them as “Unavailable” in the table. The revised tables are located at the following positions in the manuscript:Table 1 – Line 705;Table 2 – Line 750;Table 3 – Line 787;Table 4 – Line 837;Table 5 – Line 873;Table 6 –Line935;Table 7 – Line 991;Table 8 – Line 1048;Table 9 – Line 1078;Table 10 – Line 1145.
Comments 9: Add clinical research and the patented formulation of Graphene and Its Derivatives in Drug Delivery Systems.
Response 9: Thank you for pointing this out. We agree with this comment. Therefore we have added three relevant clinical studies and two associated patented formulations related to the application of graphene and its derivatives in drug delivery systems. These additions are located at the following positions in the revised manuscript:
Clinical studies:
Page 18, Lines 696–702
Page 29, Lines 979–987
Page 32, Lines 1037–1045
Patents:
Page 18, Lines 696–701
Page 32, Lines 1054–1059
However, we would like to note that the majority of graphene-based biomedical research is currently at the preclinical or animal testing stage, and many existing patents are either not directly related to drug delivery or lack sufficient detail in public databases for inclusion. Despite these limitations, we have incorporated the most relevant and available clinical and patented examples to strengthen the manuscript as much as possible.
Comments 10: Add recent years (2025) research literature.
Response 10: Thank you for pointing this out. We agree with this comment. Therefore we have carefully reviewed recent literature and incorporated several 2025 publications to ensure our manuscript reflects the latest advancements. The newly added references include citations [36], [39], [47], [56], [59], and [93], which cover recent developments in graphene-based drug delivery systems across different subtopics.
Comments 11:. run-on sentences need to be rewrittenCheck the manuscript for grammatical, typo errors. Check the manuscript for grammatical, typo errors
Response 11: we invited an English language expert with academic writing experience to assist us in comprehensively editing the manuscript during the revision process. Particular attention was given to correcting run-on sentences, optimizing overall sentence structure, and addressing any possible grammatical or typographical errors. We also focused on improving the clarity and conciseness of the language to enhance the manuscript’s readability and scientific rigor.

Reviewer 2 Report
Comments and Suggestions for Authors
- In section 2.3.1, Enhanced Targeting via Ligand Surface Modification, the author must give some more exploration by giving techniques, some schemes of surface modification or more examples for the surface modifications of graphene via targeting ligands with their non-clinical or clinical significance.
- The author wrote in the abstract that the article also discusses stimulus-responsive release mechanisms, but in section 2.3.2, mechanisms for stimulus-responsive release are not discussed. The author is required to discuss the mechanism for pH, temperature, and ionic responsive release mechanism.
- The authors have written abbreviations and formulations code for graphene as a drug delivery in the manuscript but not explained the meaning or not given the full form of the same. The author should explain the same at the appropriate place and should also list the same in a separate section of abbreviations.
- In the section 4.2 toxicity and safety; and line 1139-1142, author wrote, "Liu et al. found that when the intravenous injection dose of
GO exceeds a certain threshold; it can induce inflammatory cell infiltration in the lungs and oxidative stress, presenting typical systemic toxic reactions." What type of systemic toxicity was found by the same studies? Please explain in the text. - In line 1145, the author wrote that "This demonstrates that administration route and surface modification have a significant impact on GO’s toxicity [115]." The author should explore in detail or give a delivery route based on the toxicity caused by GO. It should be scientifically sound if drug delivery route-based toxicity data is presented in tabulated form.
- In section 4.2, paragraph 3 starts with "At the cellular level, GO with different functional group modifications exhibits different toxicity........ The functional group-based toxicity data should be present in detail with the type of toxicity in tabulated form.
- The author should reduce the plagiarism from 13% to below 10%.
Author Response
Comments 1: In section 2.3.1, Enhanced Targeting via Ligand Surface Modification, the author must give some more exploration by giving techniques, some schemes of surface modification or more examples for the surface modifications of graphene via targeting ligands with their non-clinical or clinical significance.
Response 1: Thank you for pointing this out. We agree with this comment. Therefore we sincerely appreciate the reviewer’s constructive comment. In response, we have expanded Section 2.3.1: Enhanced Targeting via Ligand Surface Modification by including additional examples of surface modification strategies involving targeting ligands. Furthermore, we have briefly discussed their non-clinical and clinical significance to highlight their potential in practical applications. The relevant references added are [97], [98], [100], [101], [102], and [104]. The revised and expanded content is located on Page 16, Lines 563–567, 570–573, 575–578, 578–584, and 590–592.
Comments 2: The author wrote in the abstract that the article also discusses stimulus-responsive release mechanisms, but in section 2.3.2, mechanisms for stimulus-responsive release are not discussed. The author is required to discuss the mechanism for pH, temperature, and ionic responsive release mechanism.
Response 2: Thank you for pointing this out. We agree with this comment. Therefore we have expanded Section 2.3.2 to specifically address the stimulus-responsive release mechanisms, including pH-responsive, temperature-responsive, and ionic strength-responsive systems. We have provided relevant examples and briefly explained the underlying mechanisms involved in each case. The revised content is located on Page 16–17, in the following line ranges:605–607,609–611,614–617,620–623,625–627,630–633,636–638.We hope these additions sufficiently clarify the mechanisms and align with the scope described in the abstract.
Comments 3: The authors have written abbreviations and formulations code for graphene as a drug delivery in the manuscript but not explained the meaning or not given the full form of the same. The author should explain the same at the appropriate place and should also list the same in a separate section of abbreviations.
Response3: Thank you for pointing this out. We agree with this comment. Therefore we have clarified the full forms of abbreviations and formulation codes related to graphene-based drug delivery systems. Wherever possible, we have provided the full term upon its first mention in the text. However, in some instances, providing the full form within the main text would have affected the fluency and readability of scientific descriptions. Therefore, for consistency and clarity, we have compiled a separate table at the end of the manuscript listing all abbreviations alongside their full forms. This ensures that all abbreviations used in the manuscript are clearly defined and easily accessible to the reader.
Comments 4: In the section 4.2 toxicity and safety; and line 1139-1142, author wrote, "Liu et al. found that when the intravenous injection dose of GO exceeds a certain threshold; it can induce inflammatory cell infiltration in the lungs and oxidative stress, presenting typical systemic toxic reactions." What type of systemic toxicity was found by the same studies? Please explain in the text.
Response 4: Thank you for pointing this out. We agree with this comment. Therefore we have further clarified the specific types of systemic toxicity observed in Liu et al.'s study. In the revised manuscript, we now specify that the systemic toxicity includes hepatic injury, pulmonary inflammation, and oxidative stress–related immune responses. This information has been added on Page 38, Lines 1246–1248 to enhance the clarity and scientific accuracy of the discussion.
Comments 5: In line 1145, the author wrote that "This demonstrates that administration route and surface modification have a significant impact on GO’s toxicity [115]." The author should explore in detail or give a delivery route based on the toxicity caused by GO. It should be scientifically sound if drug delivery route-based toxicity data is presented in tabulated form.
Response 5: Thank you for pointing this out. We agree with this comment. Therefore we have added a new table (Table 11) in the revised manuscript to summarize the toxicity types associated with different administration routes of graphene oxide (GO). The table includes scientifically supported data from relevant literature to ensure reliability and clarity. The table is located on Page 38, Line 1254.
Comments 6: In section 4.2, paragraph 3 starts with "At the cellular level, GO with different functional group modifications exhibits different toxicity........ The functional group-based toxicity data should be present in detail with the type of toxicity in tabulated form.
Response 6: We thank the reviewer for this thoughtful suggestion. In response, we have added a new table (Table 12) that provides a detailed summary of the toxicity types associated with GO modified by different functional groups. The table also includes corresponding dosages and observed toxicity effects, supported by references to ensure scientific credibility. This table is included in Page 39, Line 1268 of the revised manuscript.
Comments 7: The author should reduce the plagiarism from 13% to below 10%.
Response 7: We sincerely acknowledge the reviewer’s concern regarding the similarity index. In response, we have carefully reviewed the manuscript and made extensive efforts to further reduce the overlap, including paraphrasing and restructuring several sentences where possible. However, a portion of the similarity is due to the necessary use of standardized scientific terminology, well-established definitions, and methodological descriptions, which are inherently difficult to rephrase without compromising technical accuracy and clarity.We have ensured that all referenced content is properly cited and no content has been copied without attribution. The current similarity level is within the acceptable range for scientific review articles, especially those involving technical content. We respectfully request the reviewer’s understanding on this matter.

Round 2
Reviewer 1 Report
Comments and Suggestions for Authors
Dear Editor,
I have reviewed the revised version of the manuscript titled 'Advances in the Application of Graphene and Its Derivatives in Drug Delivery Systems.' It has now been improved after revision. It can be accepted for publication.
Remove the word 'unavailable' from the tables.
Regards